# TabRep: Training Tabular Diffusion Models with a Simple and Effective Continuous Representation

**Jacob Si**[1], **Zijing Ou**[1*], **Mike Qu**[2*], **Zhengrui Xiang**[1*], **Yingzhen Li**[1]
*Imperial College London*[1]
*Columbia University*[2]
*{y.si23, yingzhen.li}@imperial.ac.uk*

**Reviewed on OpenReview:** *https://openreview.net/forum?id=yRbtFEh2OP*

## Abstract

Diffusion models have been the predominant generative model for tabular data generation. However, they face the conundrum of modeling under a separate versus a unified data representation. The former encounters the challenge of jointly modeling all multi-modal distributions of tabular data in one model. While the latter alleviates this by learning a single representation for all features, it currently leverages sparse suboptimal encoding heuristics and necessitates additional computation costs. In this work, we address the latter by presenting TABREP, a tabular diffusion architecture trained with a unified continuous representation. To motivate the design of our representation, we provide geometric insights into how the data manifold affects diffusion models. The key attributes of our representation are composed of its density, flexibility to provide ample separability for nominal features, and ability to preserve intrinsic relationships. Ultimately, TABREP provides a simple yet effective approach for training tabular diffusion models under a continuous data manifold. Our results showcase that TABREP achieves superior performance across a broad suite of evaluations. It is the first to synthesize tabular data that exceeds the downstream quality of the original datasets while preserving privacy and remaining computationally efficient. Code: https://github.com/jacobyhsi/TabRep.

## 1 Introduction

Tabular data are ubiquitous in data ecosystems of many sectors such as healthcare and finance (Clore et al., 2014; Moro et al., 2012; Si et al., 2024). These industries utilize tabular data generation for many practical purposes, including data augmentation (Jolicoeur-Martineau et al., 2024), privacy-preserving machine learning (Xu et al., 2021), and handling sparse, imbalanced datasets (Onishi & Meguro, 2023; Sauber-Cole & Khoshgoftaar, 2022). Unlike homogeneous data modalities such as images or text, one notable characteristic inherent to tabular data is feature heterogeneity (Liu et al., 2023). Tabular data often contain mixed feature types, ranging from (dense) continuous features to (sparse) categorical features.

Recently, the best-performing tabular generative models are based on diffusion models (Ho et al., 2020b; Song & Ermon, 2020). For instance, (Lee et al., 2023; Kotelnikov et al., 2023; Shi et al., 2024b) leverage continuous (Ho et al., 2020b; Song & Ermon, 2020) and discrete diffusion (Hoogeboom et al., 2021; Shi et al., 2024a) to generate tabular data. However, these methods are designed to jointly optimize a multimodal continuous-discrete data representation that complicates the training process. On the other hand, while solutions that optimize a unified continuous representation circumvent this by learning a single representation for all features, they rely on encoding heuristics such as one-hot (Kim et al., 2022) or learned latent embeddings using a $\beta$-VAE (Zhang et al., 2023). Unfortunately, one-hot encoding for categorical variables leads to sparse representations in high dimensions, where generative models are susceptible to underfitting (Krishnan et al., 2017; Poslavskaya & Korolev, 2023). Furthermore, using a latent embedding space (Zhang et al., 2023)

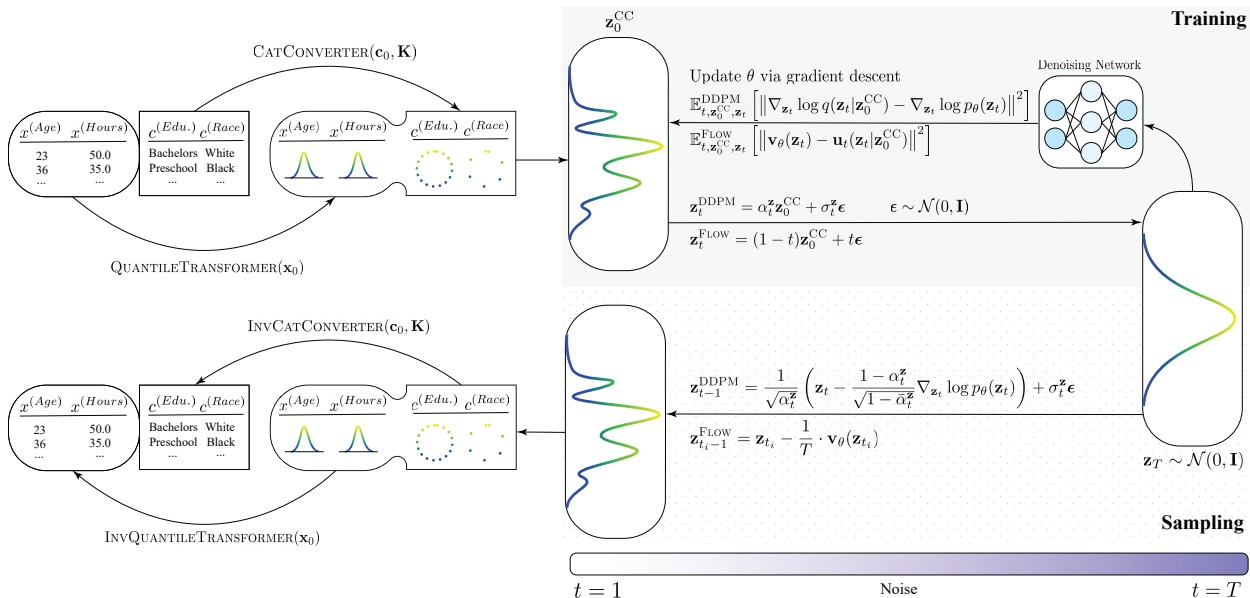

Figure 1: The TABREP Architecture. TABREP transforms and unifies the data space under a continuous regime via the our representation. A diffusion or flow matching process is trained to optimize the denoising network. Once training is completed, samples can be generated through a reverse denoising process before inverse transforming back into their original data representation.

requires training the additional VAE embedding model (Higgins et al., 2017; Kingma & Welling, 2013) that requires extra computation while being dependent on the quality of the latent space.

Since diffusion models rely on continuous transformations of denoising score-matching, or invertible mappings between data and latent spaces, designing an effective data representation is key for high-performing diffusion models (Bengio et al., 2014). In this work, we propose TABREP, a simple and effective continuous representation for training tabular diffusion models. We introduce geometric insights to understand the properties of a continuous data manifold for tabular diffusion models. Then, we craft an abstract and useful representation, that enables diffusion models like DDPM (Ho et al., 2020a) and Flow Matching (Lipman et al., 2022) to capture the posterior distribution efficaciously. TABREP's representation maintains a dense two-dimensional representation, circumventing the curse of dimensionality. It is also well-suited to model the cardinality of nominal features, ensuring ample separability to distinguish between different categories. Lastly, coupling the cyclical nature of our representation with the separability it offers enables it to facilitate ordinal features. These attributes present desirable characteristics that make it easier for diffusion models to extract meaningful information from a unified continuous tabular data representation. We conduct comprehensive experiments against tabular diffusion baselines across various datasets and benchmarks. The results showcase that TABREP consistently outperforms these baselines in quality, privacy preservation, and computational cost, indicating our superior capabilities to generate tabular data. Our architecture is in Figure 1.

## 2 Related Work

The latest tabular diffusion models have made considerable progress compared to previous generative models such as VAEs (Xu et al., 2019) and GANs (Xu et al., 2019). This included STaSy (Kim et al., 2022), which employed a score-matching diffusion model paired with self-paced learning and fine-tuning to stabilize the training process, and CoDi (Lee et al., 2023), which used separate diffusion schemes for categorical and numerical data along with interconditioning and contrastive learning to improve synergy among features. TabDDPM (Kotelnikov et al., 2023) presented a similar diffusion scheme compared to CoDi and showed that the simple concatenation of categorical and numerical data before and after denoising led to improvements in performance. TabSYN (Zhang et al., 2023) is a latent diffusion model that transformed features into a unified embedding via a $\beta$-VAE (Kingma & Welling, 2013) before applying EDM diffusion (Karras et al.,

2022) to generate synthetic data. Schröder et al. (2024) proposed TabED, the first successful attempt that applied a deep energy based model via energy discrepancy (Schröder et al., 2023). CDTD (Mueller et al., 2025) combines score matching and score interpolation to enforce a unified continuous noise distribution for both continuous and categorical features but different benchmarks are used to perform evaluation. TabDiff, couples DDPM (Ho et al., 2020b) and Discrete Masked Diffusion (Shi et al., 2024a; Sahoo et al., 2024) to synthesize tabular data. Recent works have also explored using Large Language Models (LLMs) for tabular data generation and augmentation, through techniques such as fine-tuning and in-context prompting to capture feature-label dependencies, augment scarce samples, and oversample minority classes (Nguyen et al., 2024; Seedat et al., 2024; Yang et al., 2024). While these LLM-based methods demonstrate impressive few-shot generalization, they remain constrained by limited context length and token overhead, restricting their applicability to large and complex data regimes (Yang et al., 2025; Touvron et al., 2023). In contrast, diffusion/flow-based generators such as TabRep scale easily to large datasets and allow for direct, explicit modeling over the joint distribution of mixed-type features.

## 3 Tabular Diffusion Models

Unlike images that contain only a single data type, the heterogeneous nature of tabular data requires an intricate mechanism to model continuous and discrete features. We begin with the following preambles where we denote a tabular dataset as $\mathcal{D} = \{\mathbf{z}^{(i)}\}_{i=1}^N = \{[\mathbf{x}^{(i)}, \mathbf{c}^{(i)}]\}_{i=1}^N$, where $N$ represents the number of samples (rows). Each sample consists of continuous (numerical) features $\mathbf{x}^{(i)} \in \mathbb{R}^{D_{\text{cont}}}$ and discrete (categorical) features $\mathbf{c}^{(i)} \in \prod_{j \in \{1,\dots,D_{\text{cat}}\}} \{1,\dots,K_j\}$, where $D_{\text{cont}}$ is the number of continuous features, $D_{\text{cat}}$ is the number of categorical features, and $K_j$ is the number of unique categories for the $j$-th categorical feature.

### 3.1 Tabular Diffusion Framework

Traditional Diffusion Models (Ho et al., 2020b; Song & Ermon, 2020; Sohl-Dickstein et al., 2015) operate under a Markovian noising and denoising process where the forward noising iteratively corrupts the data distribution into a Gaussian distribution via Gaussian convolution, and the reverse denoising learns to transform Gaussian noise back into meaningful data samples. Due to the multi-modality of tabular data, tabular diffusion models have been designed to handle each tabular column as its innate continuous and discrete data type. The joint continuous-discrete forward diffusion process is defined as:

$$q(\mathbf{z}_t|\mathbf{z}_0) = q(\mathbf{x}_t|\mathbf{x}_0) \cdot q(\mathbf{c}_t|\mathbf{c}_0). \tag{1}$$

The reverse joint diffusion process parameterized by $\theta$ is learned and given by:

$$p_\theta(\mathbf{z}_{t-1}|\mathbf{z}_t) = p_\theta(\mathbf{x}_{t-1}|\mathbf{x}_t, \mathbf{c}_t) \cdot p_\theta(\mathbf{c}_{t-1}|\mathbf{x}_t, \mathbf{c}_t). \tag{2}$$

At a high level, the latest state-of-the-art tabular diffusion models (Shi et al., 2024b; Kotelnikov et al., 2023; Lee et al., 2023) are trained on a separate continuous-discrete data representation via continuous Gaussian diffusion (Ho et al., 2020b; Karras et al., 2022) and discrete diffusion (Shi et al., 2024a; Hoogeboom et al., 2021).

**Gaussian Diffusion of Continuous Features**. The diffusion process of continuous features can be formulated using discrete-time Gaussian diffusion (Ho et al., 2020b; Nichol & Dhariwal, 2021); the noising process is described as:

$$q(\mathbf{x}_t|\mathbf{x}_{t-1}) = \mathcal{N}(\mathbf{x}_t|\sqrt{\alpha_t^{\mathbf{x}}}\mathbf{x}_{t-1}, (1-\alpha_t^{\mathbf{x}})\mathbf{I}), \tag{3}$$

where $\alpha_t^{\mathbf{x}}$ is a hyperparameter to control the magnitude of the noising process. Per Tweedie's Lemma (Efron, 2011; Robbins et al., 1992), the mean of the denoising distribution can be obtained by the score of the distribution:

$$\mathbb{E}_q[\mathbf{x}_{t-1}|\mathbf{x}_t] = \frac{1}{\sqrt{\alpha_t^{\mathbf{x}}}} \left( \mathbf{x}_t - \frac{1-\alpha_t^{\mathbf{x}}}{\sqrt{1-\bar{\alpha}_t^{\mathbf{x}}}} \nabla_{\mathbf{x}_t} \log q(\mathbf{x}_t) \right), \tag{4}$$

in which we can use the approximated score function $\nabla_{\mathbf{x}} \log p_\theta(\mathbf{x}) \approx \nabla_{\mathbf{x}} \log q(\mathbf{x})$ learned using denoising score matching (Song & Ermon, 2020; Vincent, 2011). This yields the denoising loss function for continuous features:

$$\mathcal{L}_{\mathbf{x}} = \mathbb{E}_{t,\mathbf{x}_0,\mathbf{x}_t,\mathbf{c}_t} \left[ \|\nabla_{\mathbf{x}_t} \log q(\mathbf{x}_t|\mathbf{x}_0) - \nabla_{\mathbf{x}_t} \log p_\theta(\mathbf{x}_t, \mathbf{c}_t)\|^2 \right]. \tag{5}$$

**Discrete Diffusion of Categorical Features**. Generating categorical features can be achieved using discrete-time diffusion processes tailored to handle discrete data (Hoogeboom et al., 2021; Campbell et al., 2022). The noising process for categorical variables is defined by transitioning between categorical states over discrete timesteps. The reverse process aims to learn the transition probabilities.

**Unified Continuous Diffusion**. Training diffusion models on a separate continuous-discrete data representation comes with the challenge of joint optimization. An alternative is to model a unified continuous data representation that circumvents these challenges. STASY (Kim et al., 2022) utilizes a naive one-hot encoding representation to unify the data space but necessitates additional computationally heavy tricks to improve its performance. It trains a diffusion model on a subset of data before gradually extending to the whole training set. Thus, in addition to the diffusion model parameter, $\theta$, the model is required to learn a selection importance vector, $\mathbf{v} \in [0, 1]$ through an alternative convex search (Bazaraa et al., 1993). The search solves the loss function by iteratively optimizing between $\theta$ and $\mathbf{v}$ while also requiring a self-paced regularizer, $r(.)$, that requires fine-tuning. Another tabular diffusion method, TABSYN (Zhang et al., 2023), unifies the data space by transforming discrete features using a learned $\beta$-VAE latent embedding. The architecture of the VAE consists of a tokenizer that encodes discrete features into a one-hot vector, then parameterizes each category as a learnable vector learned by a Transformer (Vaswani, 2017) encoder. Likewise to STASY, TABSYN is also computationally expensive since it relies on training a separate VAE before the diffusion process to obtain latent codes. Additionally, the quality of synthetic data is highly dependent on the quality of the latent space. In the following section, we show that leveraging our simple and effective continuous data representation to train tabular diffusion models is essential in synthesizing high-quality tabular data.

## 4 Method

Tabular data undergo preprocessing preceding the diffusion generative process, and transforming them into intelligible representations streamlines and improves the training of diffusion models.

### 4.1 Geometric Implications on the Data Representation

In traditional deep learning (Goodfellow et al., 2016), a sparse representation suffers from the curse of dimensionality (Bellman, 1957), where the feature space grows exponentially with the number of categories, reducing model generalization (Krishnan et al., 2017; Poslavskaya & Korolev, 2023). While maintaining a dense representation is key, representation learning also (Bengio et al., 2013; LeCun et al., 2015) emphasizes the importance of separability, enabling the neural network to learn decision boundaries in the continuous embedding space. In the following excerpt, we find that balancing both *density* and *separability* while incorporating *order* is essential for unified tabular diffusion.

**Density**. Diffusion models, which learn to generate data through iterative perturbations and reconstructions of noise, encounter geometric challenges when applied to high-dimensional, sparse representations of categorical features. In this discussion, we use the sparse one-hot representation as an exemplar to provide geometric insights into these challenges.

Let $\{e_1, e_2, \ldots, e_K\} \subset \mathbb{R}^K$ denote the set of one-hot vectors, where

$$e_k = (0, \ldots, 0, \underbrace{1}_{k\text{-th entry}}, 0, \ldots, 0), \tag{6}$$

represents the $k$-th category. For any point $x \in \mathbb{R}^K$, let

$$d_k(x) = \|x - e_k\|_2, \tag{7}$$

denote the Euclidean distance between $x$ and $e_k$. On the data manifold, there exist "singular" (Wikipedia contributors, 2025) points where the vector field is hard to learn for diffusion models with Gaussian transitions. We demonstrate this with a uniform $K$-categorical distribution. Specifically, we define:

**Definition 4.1** (*n*-singular point). *A point $x \in \mathbb{R}^K$ is an n-singular point if there exists a subset $\mathcal{S} \subseteq \{1, \ldots, K\}$ with $|\mathcal{S}| = n$ such that:*

1. $d_k(x) = d_{k'}(x), \quad \forall k, k' \in \mathcal{S}$.

2. $d_k(x) \neq d_m(x), \quad \forall k \in \mathcal{S}, \forall m \notin \mathcal{S}$.

An $n$-singular point can be extended as a *minimal $n$-singular point* if it satisfies Definition 4.1, and minimizes the Euclidean distance to all one-hot vectors in $\mathcal{S}$: $\min_x \|x - e_k\|_2 \ \forall k \in \mathcal{S}$. Hence, the minimal $n$-singular point is given by:

$$x_{\mathcal{S}}^{(n)} = \frac{1}{n} \sum_{k \in \mathcal{S}} e_k, \tag{8}$$

that corresponds to the centroid of the $n$ one-hot vectors.

**Definition 4.2** ($n$-singular hyperplane). *For each minimal $n$-singular point, there exists an $n$-singular hyperplane that is comprised of the set of all $n$-singular points associated with its respective $n$ one-hot vectors in $\mathcal{S}$. Formally, it is defined as:*

$$H_{\mathcal{S}} = \{ x \in \mathbb{R}^K \mid d_k(x) = d_{k'}(x), \quad \forall k, k' \in \mathcal{S} \}, \tag{9}$$

where $H_{\mathcal{S}}$ is an affine subspace in $\mathbb{R}^K$ of dimension $\dim(H_{\mathcal{S}}) = K - |\mathcal{S}| + 1$. The hyperplane spans the corresponding minimal $n$-singular point $x_{\mathcal{S}}^{(n)}$ and non-minimal $n$-singular points. For each $n < K$, there are $\binom{K}{n}$ distinct $n$-singular hyperplanes, one for each subset $\mathcal{S}$ of size $n$. Across all $2 \leq n \leq K$, the number of minimal $n$-singular points on the probability simplex scales combinatorially: $\sum_{n=2}^{K} \binom{K}{n} = 2^K - (K+1)$. Thus, each minimal singular point carries the additional complexity of a continuous singular hyperplane.

Diffusion models rely on gradients derived from a learned vector field to denoise data iteratively (Ho et al., 2020a; Lipman et al., 2022). For regions in proximity to the singular hyperplanes, learning the gradients of diffusion models suffers from high variance due to conflicting directions arising from equidistant one-hot points. To analyze how diffusion models behave on categorical manifolds, we study the forward diffusion process applied to one-hot encoded features. In diffusion-based generative modeling, each data point is progressively perturbed by Gaussian noise according to a transition kernel $p_t(x|e_k) = \mathcal{N}(x|\alpha_t e_k, \sigma_t^2 I)$. When categorical variables are represented as one-hot vectors, these Gaussians are centered at the simplex vertices $e_k$. Understanding how the noisy samples $x$ from these distributions overlap—and how their corresponding score functions behave—reveals the geometric instability that arises when the diffusion process operates on sparse categorical spaces. This motivates our study of singular regions where multiple one-hot categories become equidistant to a noisy sample. In the following, we show that the variance of the conditional score function increases asymptotically with the degree of $n$-singular points.

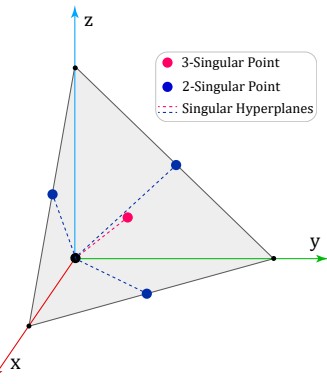

Figure 2: Singular Regions in a 3D One-Hot Setting.

**Theorem 4.1** (Variance of Conditional Score Function). *Assume $x$ is a noisy observation from a Gaussian centered at a weighted one-hot vector $\alpha_t e_k \in \mathbb{R}^K$. We can define the forward diffusion process as: $p_t(x|e_k) = \mathcal{N}(x|\alpha_t e_k, \sigma_t^2 I)$. We derive the variance of the conditional score function evaluated at a minimal $n$-singular point as:*

$$\mathrm{Var}(g|x) = \frac{\alpha_t^2}{\sigma_t^4} \frac{n-1}{n}, \tag{10}$$

where we define the conditional and expected score as $g$ and $\bar{g}$. See Appendix A.1 for proofs using a uniform and categorical prior. We find that near a minimal $n$-singular point, $x$, the posterior-weighted variance of the conditional score function is strictly positive and increases asymptotically with $n$. In contrast, at a non-singular point, the posterior leans towards $e_{k^*}$, leading the score variance to approach zero.

In Figure 2, we depict a three-dimensional setting of the one-hot representation. As illustrated, a minimal 3-singular point occurs at the (red) centroid, $\left(\frac{1}{3}, \frac{1}{3}, \frac{1}{3}\right)$. Along with the non-minimal 3-singular points, these points form a (red-dashed) 1-dimensional singular hyperplane $H_{\{1,2,3\}}$ perpendicular to the centroid of the probability simplex formed by $e_1, e_2, e_3$. Similarly, there are $\binom{3}{2}$ 2-singular (blue) points accompanied by

their respective (blue-dashed) hyperplanes $H_{\{1,2\}}$, $H_{\{1,3\}}$, and $H_{\{2,3\}}$, each of which is a 2-dimensional affine subspace (plane). These singular hyperplanes pose more difficulty for diffusion models to learn effectively, especially when $K$ is large.

**Separability and Order**. A sparse representation naturally accommodates separability for nominal features, enabling one-hot to assign each category to each dimension. However, higher-dimensional spaces introduce higher-order singular hyperplanes that complicate training. This presents a density-separability trade-off. In our experiments (Table 3), we discover that *sparse representations has a significant impact on harming diffusion generative performance*. Thus, our initial goal is to reduce the dimensionality when designing our representation.

Methods like Analog Bits (Chen et al., 2022) admit a similar idea of shrinking data dimension. Hence, a potential concern of a dense representation is to retain their ability of representing nominal features. However, the notion of separability (Bengio et al., 2014) demonstrates that nominal features can still be effectively encoded by dense representations, provided they are *sufficiently separated within the embedding space*. This perspective aligns with learned entity embeddings (Guo & Berkhahn, 2016), which demonstrate that nominal categories—despite lacking inherent order—can be effectively represented as low-dimensional embeddings. However, for datasets with a large presence of ordinal features such as "Education" in Adult and "Day" in Beijing, our experiment in Appendix B.2, Table 4 highlights that *order is an important factor for ordinal features*. Therefore, designing a dense representation that is separable and capable of encoding ordinal structure is critical for encoding both nominal and ordinal features.

## 4.2 TabRep Architecture

**CatConverter**. Inspired by the discrete Fourier transform (DFT) (Oppenheim, 1999; Bracewell & Kahn, 1966), we draw on the concept of roots of unity to design a continuous representation for diffusion models to generate categorical variables. We refer to our representation as the CatConverter. In harmonic analysis, the DFT maps signals into the frequency domain using complex exponentials, where the $K$-th roots of unity represent equally spaced points on the unit circle, given by phases:

$$\theta_k = \frac{2\pi k}{K_j} \quad \text{for } k = 0, 1, \dots, K_j - 1, \tag{11}$$

for each $j$ category and $i$ sample. Analogously, we treat a categorical feature $c_j^{(i)}$ with $K_j$ distinct values as selecting one of these $K_j$ points on the unit circle. Each category is thus mapped to a unique phase, and we represent it using the real and imaginary components of the corresponding complex exponential:

$$\text{CatConverter}(c_j^{(i)}, K_j) = \left[ \cos\left( \frac{2\pi c_j^{(i)}}{K_j} \right), \sin\left( \frac{2\pi c_j^{(i)}}{K_j} \right) \right], \tag{12}$$

where $\text{CatConverter}() \in \mathbb{R}^{2 \cdot D_{\text{CAT}}}$. Viewing a categorical entry as a discrete harmonic index enables us to embed the category in a two-dimensional phase space. This viewpoint retains the geometric insight from the DFT—the roots of unity form a symmetric and uniform structure on the unit circle—providing a smooth, dense, and geometry-aware representation for categorical variables.

In Table 1, CatConverter offers a dense 2D representation relative to alternate categorical representations.

Table 1: Categorical Representation Dimensions.

| REPRESENTATION | DIMENSIONS |
|---|---|
| ONE-HOT | $\sum_{j=1}^{D_{\text{CAT}}} K_j$ |
| LEARNED EMBEDDING | $d_{\text{EMB}} \cdot D_{\text{CAT}}$ |
| ANALOG BITS | $\sum_{j=1}^{D_{\text{CAT}}} \lceil \log_2(K_j) \rceil$ |
| DICTIONARY | $D_{\text{CAT}}$ |
| CATCONVERTER | $2 \cdot D_{\text{CAT}}$ |

For CatConverter, there exists one minimal $K$-singular point. Despite that, and $K$ number of 2-singular points, there are no $n$ singular points or hyperplanes for $2 < n < K$. Therefore, any $n$-singular point for $n > 2$ coincides with the minimal $K$-singular point in this 2D representation.

Next, we show that CatConverter's representation has ample separability for handling high-cardinality nominal categorical features commonly found in tabular datasets. We train a simple two-layer MLP to predict the category label from the representation. As illustrated in Figure 3, the MLP achieved perfect accuracy for up to 128 categories mapped onto a continuous phase space. Further experiments under higher cardinality settings can be found in D.4. Under high cardinality settings, the radius of CatConverter can be increased to potentially account for more categories. We leave this for future exploration.

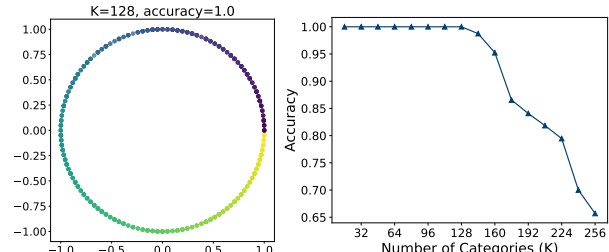

Figure 3: Separability of CatConverter. CatConverter preserves *nominal* features for up to 128 categories.

In addition to nominal features, the separability of CatConverter coupled with its circular geometry, naturally accommodates both periodic and ordinal features. This facilitates an enhanced preservation of the feature's intrinsic nature and characteristics. Note that prior to CatConverter, we introduced a static one-dimensional embedding, Dictionary (DIC), but we found that it underperformed.

**Diffusion Model**. We demonstrate that TABREP's representation is effective when modeled by either a DDPM (Ho et al., 2020b) or a Flow Matching (Lipman et al., 2022) unified continuous diffusion process. To unify the dataspace, we represent discrete variables via CatConverter($\mathbf{c}_0, \mathbf{K}$) and concatenate with continuous features forming our dataset. Our dataset can be denoted as $\{\mathbf{z}_0^{\mathrm{CC}}\} = \{[\mathbf{x}_0, \mathbf{c}_0^{\mathrm{CC}}]\}$.

For DDPM (Ho et al., 2020b), we define the forward process by progressively perturbing the data distribution using a Gaussian noise model, where the latent state at time $t$, $\mathbf{z}_t$, is computed as $\mathbf{z}_t = \alpha_t^{\mathbf{z}} \mathbf{z}_0^{\mathrm{CC}} + \sigma_t^{\mathbf{z}} \boldsymbol{\epsilon}$ with $\boldsymbol{\epsilon} \sim \mathcal{N}(0, \mathbf{I})$. The denoising model $p_\theta(\mathbf{z}_t)$ is trained to predict the posterior gradients $\nabla_{\mathbf{z}_t} \log q(\mathbf{z}_t | \mathbf{z}_0^{\mathrm{CC}})$, minimizing the weighted variance loss $\mathcal{L}_{\mathrm{TABREP\text{-}DDPM}}$:

$$\mathbb{E}_{t, \mathbf{z}_0^{\mathrm{CC}}, \mathbf{z}_t} \left[ \left\| \nabla_{\mathbf{z}_t} \log q(\mathbf{z}_t | \mathbf{z}_0^{\mathrm{CC}}) - \nabla_{\mathbf{z}_t} \log p_\theta(\mathbf{z}_t) \right\|^2 \right]. \tag{13}$$

For Flow Matching (Lipman et al., 2022), we instead define the dynamics in terms of a conditional vector field $\mathbf{u}_t(\mathbf{z}_t | \mathbf{z}_0^{\mathrm{CC}}) = \boldsymbol{\epsilon} - \mathbf{z}_0^{\mathrm{CC}}$ with $\mathbf{z}_t := (1 - t)\mathbf{z}_0^{\mathrm{CC}} + t\boldsymbol{\epsilon}$. The model learns the target field by minimizing the discrepancy between the predicted vector $\mathbf{v}_\theta(\mathbf{z}_t)$ and the ground truth $\mathbf{u}_t(\mathbf{z}_t | \mathbf{z}_0^{\mathrm{CC}})$ through the flow matching loss $\mathcal{L}_{\mathrm{TABREP\text{-}FLOW}}$:

$$\mathbb{E}_{t, \mathbf{z}_0^{\mathrm{CC}}, \mathbf{z}_t} \left[ \left\| \mathbf{v}_\theta(\mathbf{z}_t) - \mathbf{u}_t(\mathbf{z}_t | \mathbf{z}_0^{\mathrm{CC}}) \right\|^2 \right]. \tag{14}$$

At sampling time, TABREP-DDPM performs reverse diffusion by iteratively denoising $\mathbf{z}_t$ back to $\mathbf{z}_0^{\mathrm{CC}}$, while TABREP-FLOW solves a deterministic ordinary differential equation (ODE) of the vector field. Our complete training and sampling algorithms can be found in Figure 4.

**InvCatConverter**. During sampling, the model outputs continuous representations $\mathbf{c}_0^{\mathrm{CC}} = [(c_{1,\sin}^{(i)}, c_{1,\cos}^{(i)}), \ldots, (c_{D_{\mathrm{cat}},\sin}^{(i)}, c_{D_{\mathrm{cat}},\cos}^{(i)})]$ for each sample $i$, where each categorical feature $j$ is represented by its cosine and sine components in the circular embedding space. To recover the discrete category index for feature $j$ in sample $i$, we first compute the phase angle:

$$\theta_j^{(i)} = \mathrm{atan2}(c_{j,\sin}^{(i)}, c_{j,\cos}^{(i)}).$$

The recovered category index corresponds to the nearest canonical phase among the $K_j$ valid roots of unity:

$$k_j^{*(i)} = \underset{k \in \{0, \ldots, K_j^{(i)} - 1\}}{\arg\min} \left\{ \min\left( |\theta_j^{(i)} - \theta_k^*|, \ 2\pi - |\theta_j^{(i)} - \theta_k^*| \right) \right\},$$

where $\theta_k^* = \frac{2\pi k}{K_j}$ for $k = 0, 1, \ldots, K_j - 1$. This nearest-phase projection maps each continuous sample back onto its valid discrete category. Out-of-index (OOI) cases may occur due to the stochasticity of DDPM and FM sampling; we handle these through OOI casting, assigning such cases to the 0-th category to preserve categorical validity. Further implementation details are provided in Appendix B.2.

---

**Algorithm 1** Training TABREP-DDPM/FLOW

1: DDPM:            FLOW MATCHING:
2: **while** not converged **do**
3:     Sample $\mathbf{z}_0 = [\mathbf{x}_0, \mathbf{c}_0] \sim p(\mathbf{z})$
4:     Encode $\mathbf{c}_0^{\text{CC}} \leftarrow \text{CatConverter}(\mathbf{c}_0, \mathbf{K})$
5:     Encode $\mathbf{x}_0 \leftarrow \text{QUANTILETRANSFORMER}(\mathbf{x}_0)$
6:     $\mathbf{z}_0^{\text{CC}} \leftarrow \text{CONCAT}(\mathbf{x}_0, \mathbf{c}_0^{\text{CC}})$
7:     Sample $t \sim \text{Uniform}(\{1, \dots, T\})$
8:     Sample noise $\boldsymbol{\epsilon} \sim \mathcal{N}(0, \mathbf{I})$
9:     Compute $\mathbf{z}_t = \alpha_t^{\mathbf{z}} \mathbf{z}_0^{\text{CC}} + \sigma_t^{\mathbf{z}} \boldsymbol{\epsilon}$           Compute $\mathbf{z}_t = (1-t)\mathbf{z}_0^{\text{CC}} + t\boldsymbol{\epsilon}$

                                                 Define $\mathbf{u}_t(\mathbf{z}_t|\mathbf{z}_0^{\text{CC}}) = \boldsymbol{\epsilon} - \mathbf{z}_0^{\text{CC}}$

    Compute $\mathcal{L}_{\text{TABREP-DDPM}} =$             Compute $\mathcal{L}_{\text{TABREP-FLOW}} =$

$\mathbb{E}_{t, \mathbf{z}_0^{\text{CC}}, \mathbf{z}_t} \left[ \left\| \nabla_{\mathbf{z}_t} \log q(\mathbf{z}_t|\mathbf{z}_0^{\text{CC}}) - \nabla_{\mathbf{z}_t} \log p_\theta(\mathbf{z}_t) \right\|^2 \right]$    $\mathbb{E}_{t, \mathbf{z}_0^{\text{CC}}, \mathbf{z}_t} \left[ \left\| \mathbf{v}_\theta(\mathbf{z}_t) - \mathbf{u}_t(\mathbf{z}_t|\mathbf{z}_0^{\text{CC}}) \right\|^2 \right]$

    Update $\theta$ via gradient descent for $\nabla_\theta \mathcal{L}_{\text{TABREP-DDPM}}$ and $\nabla_\theta \mathcal{L}_{\text{TABREP-FLOW}}$
10: **end while**

---

**Algorithm 2** Sampling TABREP-DDPM/FLOW

1: DDPM:            FLOW MATCHING:
2: Sample $\mathbf{z}_T^{\text{CC}} \sim \mathcal{N}(0, \mathbf{I})$
3: **for** $t = T, \dots, 1$ **do**
4:     Sample $\boldsymbol{\epsilon} \sim \mathcal{N}(0, \mathbf{I})$ if $t > 1$,          Discretize time $t_i = i/T$,
    else $\boldsymbol{\epsilon} = 0$                             for $i = T, T-1, \dots, 1$

    $\mathbf{z}_{t-1} = \frac{1}{\sqrt{\alpha_t^{\mathbf{z}}}} \left( \mathbf{z}_t - \frac{1-\alpha_t^{\mathbf{z}}}{\sqrt{1-\bar{\alpha}_t^{\mathbf{z}}}} \nabla_{\mathbf{z}_t} \log p_\theta(\mathbf{z}_t) \right) + \sigma_t^{\mathbf{z}} \boldsymbol{\epsilon}$    $\mathbf{z}_{t_i - 1} = \mathbf{z}_{t_i} - \frac{1}{T} \cdot \mathbf{v}_\theta(\mathbf{z}_{t_i})$,

                                                 via Euler ODE Solver
5: **end for**
6: Output $\mathbf{z}_0^{\text{CC}} = [\mathbf{x}_0, \mathbf{c}_0^{\text{CC}}]$
7: Decode $\mathbf{c}_0 \leftarrow \text{InvCatConverter}(\mathbf{c}_0^{\text{CC}}, \mathbf{K})$
8: Decode $\mathbf{x}_0 \leftarrow \text{INVQUANTILETRANSFORMER}(\mathbf{x}_0)$
9: $\mathbf{z}_0 \leftarrow \text{CONCAT}(\mathbf{x}_0, \mathbf{c}_0)$
10: **return** $\mathbf{z}_0^{\text{CC}}$

---

Figure 4: Training and sampling algorithms of TABREP-DDPM/FLOW.

## 5 Experiments

We evaluate the performance of TABREP-DDPM and TABREP-FLOW against baselines. Our results are compared against various datasets, baselines, and benchmarks.

### 5.1 Setup

**Implementation Information**. Experimental results are obtained over an average of 20 sampling seeds using the best-validated model. Continuous features are encoded using a QuantileTransformer (Pedregosa et al., 2011). The order in which the categories of a feature are assigned in our encoding schemes is based on lexicographic ordering for simplicity. Further details are in Appendix B.

**Datasets**. We select seven datasets from the UCI Machine Learning Repository to conduct our experiments. This includes Adult, Default, Shoppers, Stroke, Diabetes, Beijing, and News which contain a mix of continuous and discrete features. Further details are in Appendix C.1.

**Baselines**. We compare our model against existing diffusion baselines for tabular generation since they are the best-performing. This includes STASY (Kim et al., 2022), CODI (Lee et al., 2023), TABDDPM (Kotelnikov et al., 2023), TABSYN (Zhang et al., 2023), and TABDIFF (Shi et al., 2024b). Further details are in Appendix C.2.

**Benchmarks**. We observe that downstream task performance measured by machine learning efficiency (MLE) typically translates to most other benchmarks. Thus, the primary quality benchmark in our main paper will

Table 2: Ablation study on TABREP's unified representation versus a separate representation.

| | AUC ↑ | | | | F1 ↑ | RMSE ↓ | |
| --- | --- | --- | --- | --- | --- | --- | --- |
| METHODS | ADULT | DEFAULT | SHOPPERS | STROKE | DIABETES | BEIJING | NEWS |
| TABDDPM | $0.910_{\pm.001}$ | $0.761_{\pm.004}$ | $0.915_{\pm.004}$ | $0.808_{\pm.033}$ | $\mathbf{0.376}_{\pm.003}$ | $0.592_{\pm.012}$ | $3.46_{\pm1.25}$ |
| TABREP-DDPM | $\mathbf{0.913}_{\pm.002}$ | $\mathbf{0.764}_{\pm.005}$ | $\mathbf{0.926}_{\pm.005}$ | $\mathbf{0.869}_{\pm.027}$ | $0.373_{\pm.003}$ | $\mathbf{0.508}_{\pm.006}$ | $\mathbf{0.836}_{\pm.001}$ |
| TABFLOW | $0.908_{\pm.002}$ | $0.742_{\pm.008}$ | $0.914_{\pm.005}$ | $0.821_{\pm.082}$ | $\mathbf{0.377}_{\pm.002}$ | $0.574_{\pm.010}$ | $0.850_{\pm.017}$ |
| TABREP-FLOW | $\mathbf{0.912}_{\pm.002}$ | $\mathbf{0.782}_{\pm.005}$ | $\mathbf{0.919}_{\pm.005}$ | $\mathbf{0.830}_{\pm.028}$ | $\mathbf{0.377}_{\pm.002}$ | $\mathbf{0.536}_{\pm.006}$ | $\mathbf{0.814}_{\pm.002}$ |

Table 3: Ablation study on categorical representations under a unified continuous data space.

| | AUC ↑ | | | | F1 ↑ | RMSE ↓ | |
| --- | --- | --- | --- | --- | --- | --- | --- |
| METHODS | ADULT | DEFAULT | SHOPPERS | STROKE | DIABETES | BEIJING | NEWS |
| ONEHOT-DDPM | $0.476_{\pm.057}$ | $0.557_{\pm.052}$ | $0.799_{\pm.126}$ | $0.797_{\pm.133}$ | $0.363_{\pm.008}$ | $2.143_{\pm.339}$ | $0.840_{\pm.020}$ |
| LEARNED1D-DDPM | $0.611_{\pm.008}$ | $0.575_{\pm.012}$ | $0.876_{\pm.028}$ | $0.743_{\pm.032}$ | $0.179_{\pm.010}$ | $0.921_{\pm.006}$ | $0.858_{\pm.010}$ |
| LEARNED2D-DDPM | $0.793_{\pm.003}$ | $0.290_{\pm.009}$ | $0.103_{\pm.011}$ | $0.850_{\pm.035}$ | $0.205_{\pm.007}$ | $0.969_{\pm.005}$ | $0.857_{\pm.023}$ |
| I2B-DDPM | $0.911_{\pm.001}$ | $0.762_{\pm.003}$ | $0.919_{\pm.004}$ | $0.852_{\pm.029}$ | $0.370_{\pm.008}$ | $0.542_{\pm.008}$ | $0.844_{\pm.013}$ |
| DIC-DDPM | $0.912_{\pm.002}$ | $0.763_{\pm.003}$ | $0.910_{\pm.004}$ | $0.824_{\pm.027}$ | $\mathbf{0.375}_{\pm.006}$ | $0.547_{\pm.008}$ | $0.851_{\pm.013}$ |
| TABREP-DDPM | $\mathbf{0.913}_{\pm.002}$ | $\mathbf{0.764}_{\pm.005}$ | $\mathbf{0.926}_{\pm.005}$ | $\mathbf{0.869}_{\pm.027}$ | $0.373_{\pm.003}$ | $\mathbf{0.508}_{\pm.006}$ | $\mathbf{0.836}_{\pm.001}$ |
| ONEHOT-FLOW | $0.895_{\pm.003}$ | $0.759_{\pm.005}$ | $0.910_{\pm.006}$ | $0.812_{\pm.129}$ | $0.372_{\pm.005}$ | $0.765_{\pm.016}$ | $0.850_{\pm.017}$ |
| LEARNED1D-FLOW | $0.260_{\pm.014}$ | $0.438_{\pm.009}$ | $0.134_{\pm.015}$ | $0.142_{\pm.033}$ | $0.184_{\pm.007}$ | $0.806_{\pm.009}$ | $0.873_{\pm.007}$ |
| LEARNED2D-FLOW | $0.126_{\pm.012}$ | $0.709_{\pm.008}$ | $0.868_{\pm.007}$ | $0.180_{\pm.030}$ | $0.177_{\pm.008}$ | $0.787_{\pm.007}$ | $0.866_{\pm.005}$ |
| I2B-FLOW | $0.911_{\pm.001}$ | $0.763_{\pm.004}$ | $0.910_{\pm.005}$ | $0.797_{\pm.027}$ | $0.372_{\pm.003}$ | $0.543_{\pm.007}$ | $0.847_{\pm.014}$ |
| DIC-FLOW | $0.912_{\pm.002}$ | $0.763_{\pm.004}$ | $0.903_{\pm.005}$ | $0.807_{\pm.028}$ | $0.376_{\pm.007}$ | $0.561_{\pm.007}$ | $0.853_{\pm.014}$ |
| TABREP-FLOW | $\mathbf{0.912}_{\pm.002}$ | $\mathbf{0.782}_{\pm.005}$ | $\mathbf{0.919}_{\pm.005}$ | $\mathbf{0.830}_{\pm.028}$ | $\mathbf{0.377}_{\pm.002}$ | $\mathbf{0.536}_{\pm.006}$ | $\mathbf{0.814}_{\pm.002}$ |

be MLE for brevity, and a privacy benchmark, membership inference attacks (MIA). The remaining fidelity benchmarks include: column-wise density (CWD), pairwise-column correlation (PCC), $\alpha$-precision, $\beta$-recall, and classifier-two-sample test (C2ST). Further details regarding benchmark information and additional results are in Appendix C.3 and D.

## 5.2 Ablation Studies

**Unified and Separate Data Representations**. We analyze the impact of training tabular diffusion models between our unified data representation and a separate data representation. In Table 2, we compare TABREP-DDPM and TABREP-FLOW to TABDDPM (Kotelnikov et al., 2023) and TABFLOW. Note that we introduce TABFLOW, by modeling continuous features using Flow Matching (Lipman et al., 2022; Liu et al., 2022) and categorical features using Discrete Flow Matching (Campbell et al., 2024) under the same separate continuous-discrete data representation as TABDDPM. Results illustrate that data synthesized by diffusion-based models using TABREP's unified representation consistently outperforms a separate representation when training diffusion and flow matching models.

**Categorical Representations**. We conduct ablation studies with respect to categorical data representations used in existing diffusion baselines to assess the performance of diffusion models under a unified data representation. This includes one-hot encoding used in (Kim et al., 2022; Lee et al., 2023; Shi et al., 2024b), learned one-dimensional and two-dimensional embeddings (Mikolov et al., 2013; Guo & Berkhahn, 2016), and Analog Bits (I2B) (Chen et al., 2022) from the discrete image diffusion domain. Additionally, we introduce an intuitive static one-dimensional embedding, Dictionary (DIC). Results in Table 3 showcase that TABREP outperforms the ablated representations in generating data under a unified data representation via diffusion models. Further details regarding the various representations are in Appendix B.1.

**Ordering of Categorical Feature**. Our CatConverter representation induces an order on the categorical features. We assess how the order influences the results. By default, we assign a lexicographic ordering to the

Table 5: AUC (classification) and RMSE (regression) scores of Machine Learning Efficiency. Higher scores indicate better performance.

| | AUC ↑ | | | | F1 ↑ | RMSE ↓ | |
| Methods | Adult | Default | Shoppers | Stroke | Diabetes | Beijing | News |
|---|---|---|---|---|---|---|---|
| Real | $0.927_{\pm.000}$ | $0.770_{\pm.005}$ | $0.926_{\pm.001}$ | $0.852_{\pm.002}$ | $0.384_{\pm.003}$ | $0.423_{\pm.003}$ | $0.842_{\pm.002}$ |
| STaSy | $0.906_{\pm.001}$ | $0.752_{\pm.006}$ | $0.914_{\pm.005}$ | $0.833_{\pm.030}$ | $0.374_{\pm.003}$ | $0.656_{\pm.014}$ | $0.871_{\pm.002}$ |
| CoDi | $0.871_{\pm.006}$ | $0.525_{\pm.006}$ | $0.865_{\pm.006}$ | $0.798_{\pm.032}$ | $0.288_{\pm.009}$ | $0.818_{\pm.021}$ | $1.21_{\pm.005}$ |
| TabDDPM | $0.910_{\pm.001}$ | $0.761_{\pm.004}$ | $0.915_{\pm.004}$ | $0.808_{\pm.033}$ | $0.376_{\pm.003}$ | $0.592_{\pm.012}$ | $3.46_{\pm1.25}$ |
| TabSYN | $0.906_{\pm.001}$ | $0.755_{\pm.004}$ | $0.918_{\pm.004}$ | $0.845_{\pm.035}$ | $0.361_{\pm.001}$ | $0.586_{\pm.013}$ | $0.862_{\pm.021}$ |
| TabDiff | $0.912_{\pm.002}$ | $0.763_{\pm.005}$ | $0.919_{\pm.005}$ | $0.848_{\pm.021}$ | $0.353_{\pm.006}$ | $0.565_{\pm.011}$ | $0.866_{\pm.021}$ |
| TabRep-DDPM | $\mathbf{0.913}_{\pm.002}$ | $0.764_{\pm.005}$ | $\mathbf{0.926}_{\pm.005}$ | $\mathbf{0.869}_{\pm.027}$ | $0.373_{\pm.003}$ | $\mathbf{0.508}_{\pm.006}$ | $0.836_{\pm.001}$ |
| TabRep-Flow | $0.912_{\pm.002}$ | $\mathbf{0.782}_{\pm.005}$ | $0.919_{\pm.005}$ | $0.854_{\pm.028}$ | $\mathbf{0.377}_{\pm.002}$ | $0.536_{\pm.006}$ | $\mathbf{0.814}_{\pm.002}$ |

Table 6: Recall Scores of MIAs. A score closer to 50% is better for privacy-preservation.

| Methods | Adult | Default | Shoppers | Stroke | Diabetes | Beijing | News |
|---|---|---|---|---|---|---|---|
| STaSy | $24.51_{\pm0.44}$ | $30.37_{\pm0.99}$ | $17.54_{\pm0.19}$ | $34.63_{\pm1.39}$ | $29.75_{\pm0.16}$ | $34.06_{\pm0.33}$ | $23.49_{\pm0.67}$ |
| CoDi | $0.05_{\pm0.01}$ | $3.41_{\pm0.53}$ | $1.36_{\pm0.45}$ | $27.32_{\pm3.50}$ | $0.00_{\pm0.00}$ | $0.40_{\pm0.09}$ | $0.04_{\pm0.02}$ |
| TabDDPM | $56.90_{\pm0.29}$ | $44.96_{\pm0.59}$ | $46.08_{\pm1.57}$ | $55.12_{\pm0.81}$ | $52.06_{\pm0.11}$ | $48.35_{\pm0.79}$ | $9.88_{\pm0.52}$ |
| TabSYN | $42.91_{\pm0.31}$ | $43.71_{\pm0.89}$ | $42.14_{\pm0.76}$ | $47.97_{\pm1.27}$ | $44.42_{\pm0.28}$ | $46.53_{\pm0.52}$ | $34.42_{\pm0.90}$ |
| TabDiff | $52.00_{\pm0.35}$ | $46.67_{\pm0.55}$ | $46.86_{\pm1.20}$ | $47.15_{\pm1.30}$ | $31.01_{\pm0.22}$ | $48.20_{\pm0.65}$ | $16.03_{\pm0.66}$ |
| TabRep-DDPM | $52.78_{\pm0.21}$ | $\mathbf{48.96}_{\pm0.41}$ | $48.16_{\pm0.90}$ | $\mathbf{50.89}_{\pm0.93}$ | $\mathbf{51.43}_{\pm0.13}$ | $\mathbf{49.50}_{\pm0.52}$ | $\mathbf{40.10}_{\pm0.55}$ |
| TabRep-Flow | $\mathbf{50.51}_{\pm0.21}$ | $47.07_{\pm0.40}$ | $\mathbf{49.19}_{\pm0.86}$ | $49.43_{\pm1.16}$ | $50.33_{\pm0.13}$ | $49.14_{\pm0.81}$ | $35.48_{\pm0.78}$ |

categorical features for simplicity, a heuristic baseline not optimized for semantic structure. However, we recognize the potential for future work to explore more principled or learnable ordering strategies.

We conducted an experiment to display the performance of lexicographic ordering vs. random ordering on the Adult and Beijing datasets. These datasets were chosen because they contain variables with a natural ordering, such as "Education" (ordinal) and "Day" (periodic). As observed by the AUC disparity in Table 4, order is an important factor in our CatConverter representation. It also implicitly highlights that CatConverter's geometry aids in preserving the inherent semantics of ordinal and cyclical categorical features.

Table 4: Ablation study on ordering of categorical feature.

| | AUC ↑ | RMSE ↓ |
| Methods | Adult | Beijing |
|---|---|---|
| TabRep-DDPM (Random) | $0.776_{\pm.002}$ | $1.050_{\pm.006}$ |
| TabRep-DDPM (Lexicographic) | $0.913_{\pm.002}$ | $0.508_{\pm.006}$ |
| TabRep-Flow (Random) | $0.807_{\pm.003}$ | $1.041_{\pm.005}$ |
| TabRep-Flow (Lexicographic) | $0.912_{\pm.002}$ | $0.536_{\pm.006}$ |

### 5.3 Baseline Performance

**Generation Quality**. We benchmark TabRep against baselines on a downstream MLE task where we determine the AUC and RMSE of XGBoost (Chen & Guestrin, 2016) for classification and regression tasks on the generated synthetic datasets. The diffusion models are trained on the training set, and synthetic samples of equivalent size are generated, which are then evaluated using the testing set. As observed in Table 5, CatConverter is effective on both DDPM and Flow Matching since TabRep-DDPM and TabRep-Flow consistently attain the best MLE performance compared to existing baselines. Additionally, our method is the first to yield performance levels greater than the Real datasets including Default, Stroke, and News.

**Privacy Preservation**. We perform membership inference attacks (MIAs) to evaluate privacy preservation by assessing the vulnerability of the methods to privacy leakage (Shokri et al., 2017). A privacy-preserving model yields recall scores of 50%, indicating that the attack is no better than random guessing. In Table 6, our method preserves privacy with MIAs scoring close to 50% for recall. Precision scores can be found in Appendix D.1 which delivers similar privacy preservation insights.

Table 7: Training and Sampling Duration.

| METHODS | TRAINING (S) | SAMPLING (S) | TOTAL (S) |
|---|---|---|---|
| STASY | 6608.84 | 12.93 | 6621.77 |
| CODI | 24039.96 | 9.41 | 24049.37 |
| TABDDPM | 3112.07 | 66.82 | 3178.89 |
| TABSYN | 2373.98 + 1084.82 | 10.54 | 3469.30 |
| TABDIFF | 5640 | 15.2 | 5655.2 |
| TABREP-DDPM | 2070.59 | 59.00 | 2130.59 |
| TABREP-FLOW | 2028.33 | 3.07 | **2031.40** |

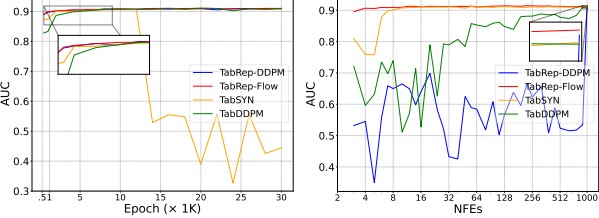

(a) AUC vs. Train Epochs  (b) AUC vs. Sample NFEs

Figure 5: Training and Sampling Efficacy.

(a) Adult

(b) Beijing

(c) Default

(d) Shoppers

(e) News

(f) Diabetes

(g) Stroke

Figure 6: Normalized Training Loss. TabRep consistently exhibits faster convergence and lower training loss, indicating improved training stability and data modeling capability.

**Generation Efficacy**. Training and Sampling TABREP is the most efficient among all baselines and does not necessitate additional computing power. We compare the generation efficacy using the Adult dataset with a balanced mix of continuous and discrete features. In Table 7, we observe that our method is the quickest to train and sample in duration. We also conducted experiments on the convergence speed for the training process. As illustrated in Figure 5a, our method converges to a high AUC earliest in the training stages. Lastly, we assess the number of function evaluations (NFEs) the models take to sample. Figure 5b indicates that TABREP-FLOW can attain its best performance as early as 8 NFEs. At 1000 NFEs, we observe that both TABREP models are the best-performing.

**Loss Stability**. We measure the training loss of TabRep by comparing the performance of CatConverter and one-hot encoding. In Figure 6, we observe that CatConverter demonstrates faster convergence, lower loss values, and a reduction in the variability of the loss.

## 6 Conclusion

In this work, we present TABREP, a simple and effective continuous representation for training tabular diffusion models. Motivated by geometric implications on the data manifold, our representation is dense, separable, and captures meaningful information for diffusion models. We conducted extensive experiments on a myriad of datasets, baselines, and benchmarks to evaluate TABREP. The results showcase TABREP's prowess in generating high-quality, privacy-preserving synthetic data while remaining computationally inexpensive. A limitation of our methodology is that we have not extensively explored a continuous embedding scheme where we perform the reverse and unify the generative space into a categorical one. Inspired by (Ansari et al., 2024), we conduct initial explorations of time series tokenization to embed continuous features. Our results are still inconclusive and left to future work.

## Declarations

**Broader Impact**. The work aims to generate synthetic tabular data that preserve statistical fidelity while maintaining privacy. Nevertheless, synthetic data can still leak sensitive information if membership inference defenses are insufficient or if generated data are misused for re-identification. Hence, practitioners should still take precautions when deploying TABREP-generated data without domain-specific privacy auditing.

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

# Appendix

## Contents

## A  Proofs

### A.1  Variance of Learning Gradients in Diffusion Models

**Theorem 4.1** (Variance of Conditional Score Function). *Assume $x$ is a noisy observation from a Gaussian centered at a weighted one-hot vector $\alpha_t e_k \in \mathbb{R}^K$. We can define the forward diffusion process as: $p_t(x|e_k) = \mathcal{N}(x|\alpha_t e_k, \sigma_t^2 I)$. We derive the variance of the conditional score function evaluated at a minimal $n$-singular point as:*

$$\mathrm{Var}(g|x) = \frac{\alpha_t^2}{\sigma_t^4} \frac{n-1}{n}, \tag{10}$$

*Proof.* **Uniform Prior**. We show that the variance of the score function at a minimal $n$-singular point increases asymptotically with respect to $n$ dimensions. Assume $x$ is a noisy observation from a Gaussian centered at a weighted one-hot vector $\alpha_t e_k \in \mathbb{R}^K$. We can define the forward diffusion process as:

$$p_t(x|e_k) = \mathcal{N}(x|\alpha_t e_k, \sigma_t^2 I) \tag{15}$$

Hence, the score function of the conditional distribution $p_t(x|e_k)$ is given by:

$$\nabla_x \log p_t(x|e_k) = \nabla_x \log \left[ \frac{1}{(2\pi\sigma_t^2)^{K/2}} \exp\left( -\frac{1}{2\sigma_t^2} ||x - \alpha_t e_k||^2 \right) \right] \tag{16}$$

$$= \nabla_x \left[ -\frac{K}{2} \log(2\pi\sigma_t^2) - \frac{1}{2\sigma_t^2} ||x - \alpha_t e_k||^2 \right] \tag{17}$$

$$= -\frac{1}{\sigma_t^2} (x - \alpha_t e_k) \tag{18}$$

In which we define it as the gradient:

$$g_k(x) := \nabla_x \log p_t(x|e_k) = -\frac{1}{\sigma_t^2}(x - \alpha_t e_k) \tag{19}$$

Assume we have a uniform prior over categories:

$$p(e_k) = \frac{1}{K} \quad \forall k \in \mathcal{S} = \{1, \ldots, K\} \tag{20}$$

Per Bayes' Rule, we compute the posterior over $e_k$:

$$q(e_k|x) = \frac{p(x|e_k)p(e_k)}{\sum_{m=1}^{K} p(x|e_m)p(e_m)} \tag{21}$$

$$= \frac{p(x|e_k)}{\sum_{m=1}^{K} p(x|e_m)} \tag{22}$$

$$= \frac{p(x|e_k)}{\sum_{k \in S} p(x|e_k) + \sum_{m \notin S} p(x|e_m)} \tag{23}$$

Since the forward process is modeled as: $p_t(x|e_k) = \mathcal{N}(x|\alpha_t e_k, \sigma_t^2 I)$, we can infer that:

$$p(x|e_k) \propto \exp\left(-\frac{1}{2\sigma_t^2}||x - \alpha_t e_k||^2\right) \tag{24}$$

Then, $\forall k \in \mathcal{S}$, the likelihood terms are identical:

$$p_t(x|e_k) = \exp\left(-\frac{d_k^2}{2\sigma_t^2}\right) := A \tag{25}$$

where $d$ is defined in Equation 7. $\forall m \notin S, d_m(x)^2 > d_k(x)^2$, thus:

$$p_t(x|e_m) = \exp\left(-\frac{d_m^2}{2\sigma_t^2}\right) \ll A \tag{26}$$

Therefore, the posterior simplifies to:

$$q(e_k|x) = \frac{p(x|e_k)}{\sum_{k \in S} p(x|e_k)} \tag{27}$$

$$\approx \frac{A}{nA} \tag{28}$$

$$= \frac{1}{n}, \quad \forall k \in \mathcal{S}. \tag{29}$$

We now compute the expected score:

$$\bar{g}(x) = \sum_{k=1}^{K} q(e_k|x) \cdot g_k(x) \tag{30}$$

$$\approx \sum_{k \in \mathcal{S}} \frac{1}{n} \cdot g_k(x) \tag{31}$$

$$= -\frac{1}{n\sigma_t^2} \sum_{k \in \mathcal{S}} (x - \alpha_t e_k) \tag{32}$$

$$= -\frac{1}{\sigma_t^2}(x - \alpha_t \bar{e}), \tag{33}$$

where $\bar{e} := \frac{1}{n} \sum_{k \in \mathcal{S}} e_k$ is the centroid of the vectors in $\mathcal{S}$. Next, compute the variance:

$$\text{Var}(g|x) = \sum_{k \in \mathcal{S}} \frac{1}{n} ||g_k(x) - \bar{g}(x)||^2 \tag{34}$$

$$= \sum_{k \in \mathcal{S}} \frac{1}{n} \left|\left| -\frac{1}{\sigma_t^2}(x - \alpha_t e_k) + \frac{1}{\sigma_t^2}(x - \alpha_t \bar{e}) \right|\right|^2 \tag{35}$$

$$= \sum_{k \in \mathcal{S}} \frac{1}{n} \cdot \frac{1}{\sigma_t^4} ||\alpha_t(\bar{e} - e_k)||^2 \tag{36}$$

$$= \frac{\alpha_t^2}{\sigma_t^4} \sum_{k \in \mathcal{S}} \frac{1}{n} ||\bar{e} - e_k||^2 \tag{37}$$

To complete the variance computation, we compute the squared distance between each one-hot vector and the centroid:

$$\|\bar{e} - e_k\|^2 = \sum_{i=1}^{K} (\bar{e}_i - e_k(i))^2 \tag{38}$$

$$= \left(\frac{n-1}{n}\right)^2 + (n-1) \cdot \left(\frac{1}{n}\right)^2 \tag{39}$$

$$= \frac{n-1}{n} \tag{40}$$

Note that for $i = k \in S$, $\bar{e}_k = \frac{1}{n}$, $e_k(k) = 1$ and for $i \neq k \in S$, $e_k(k) = 0$. Substituting into the variance:

$$\mathrm{Var}(g|x) = \frac{\alpha_t^2}{\sigma_t^4} \cdot \sum_{k \in \mathcal{S}} \frac{1}{n} \cdot \frac{n-1}{n} \tag{41}$$

$$= \frac{\alpha_t^2}{\sigma_t^4} \cdot \frac{n-1}{n} \tag{42}$$

We find that at a minimal $n$-singular point $x$, the posterior-weighted variance of the score function is strictly positive and increases asymptotically with $n$. In contrast, at a non-singular point, the posterior leans towards $e_{k^*}$:

$$q(e_{k^*}|x) \approx 1, \quad q(e_k|x) \approx 0 \quad \forall k \neq k^* \tag{43}$$

Then, the expected score becomes:

$$\bar{g}(x) = \sum_{k=1}^{K} q(e_k|x) \cdot g_k(x) \approx g_{k^*}(x) \tag{44}$$

And the variance reduces to:

$$\mathrm{Var}(g|x) = \sum_{k=1}^{K} q(e_k|x) \cdot \|g_k(x) - \bar{g}(x)\|^2 \tag{45}$$

$$\approx 1 \cdot \|g_{k^*}(x) - g_{k^*}(x)\|^2 + \sum_{k \neq k^*} 0 \cdot \|g_k(x) - g_{k^*}(x)\|^2 \tag{46}$$

$$= 0 \tag{47}$$

**Categorical Prior**. We now generalize the above result to an arbitrary categorical prior $\Pi = \{\pi_k\}_{k \in \mathcal{S}}$ over categories $\{e_k\}_{k \in \mathcal{S}}$, where $x_0 \sim \Pi$ and the forward diffusion process is defined as:

$$p_t(x|x_0) = \mathcal{N}(x|\alpha_t x_0, \sigma_t^2 I). \tag{48}$$

The marginal likelihood is then given by:

$$p_t(x) = \sum_{k \in \mathcal{S}} p_t(x|x_0 = e_k)\pi_k = \sum_{k \in \mathcal{S}} \mathcal{N}(x|\alpha_t e_k, \sigma_t^2 I)\pi_k. \tag{49}$$

Suppose we observe a noised sample $x \in \mathbb{R}^k$ at time $t$. Then the posterior probability that $x$ was generated by adding noise to $e_k$ is:

$$q_t(x_0 = e_k|x) = \frac{p_t(x|x_0 = e_k)\pi_k}{\sum_{m \in \mathcal{S}} p_t(x|x_0 = e_j)\pi_m}. \tag{50}$$

We compute the expected score at time $t$ as follows:

$$\mathbb{E}_{p_t(x)}[\nabla_x \log p_t(x)] = \mathbb{E}_{x_0 \sim \Pi}\left[\mathbb{E}_{p_t(x|x_0)}[\nabla_x \log p_t(x|x_0)]\right] \tag{51}$$

$$= \sum_{k \in \mathcal{S}} \pi_k \mathcal{N}(x|\alpha_t e_k, \sigma_t^2 I)\nabla_x \log \mathcal{N}(x|\alpha_t e_k, \sigma_t^2 I). \tag{52}$$

The score of the Gaussian is:

$$\nabla_x \log \mathcal{N}(x|\alpha_t e_k, \sigma_t^2 I) = -\frac{1}{\sigma_t^2}(x - \alpha_t e_k) = \frac{\alpha_t e_k - x}{\sigma_t^2}. \tag{53}$$

Therefore, the expected score becomes:

$$\bar{g} := \mathbb{E}_{p_t(x)}[\nabla_x \log p_t(x)] = \sum_{k \in \mathcal{S}} \pi_k \mathcal{N}(x|\alpha_t e_k, \sigma_t^2 I) \cdot \left(\frac{\alpha_t e_k - x}{\sigma_t^2}\right) \tag{54}$$

$$= C \sum_{k \in \mathcal{S}} \pi_k \exp\left(-\frac{\|x - \alpha_t e_k\|^2}{2\sigma_t^2}\right)\left(\frac{\alpha_t e_k - x}{\sigma_t^2}\right), \tag{55}$$

where $C := \frac{1}{(2\pi)^{k/2}\sigma_t^k}$.

Next, we compute the variance of the conditional score around its expectation:

$$\mathbb{E}_{x_0 \sim \Pi}\left[\mathbb{E}_{p_t(x|x_0)}\left[\|\nabla_x \log p_t(x|x_0) - \bar{g}\|^2\right]\right] = \sum_{k \in \mathcal{S}} \pi_k \mathcal{N}(x|\alpha_t e_k, \sigma_t^2 I)\left\|\frac{\alpha_t e_k - x}{\sigma_t^2} - \bar{g}\right\|^2 \tag{56}$$

$$= C \sum_{k \in \mathcal{S}} \pi_k \exp\left(-\frac{\|x - \alpha_t e_k\|^2}{2\sigma_t^2}\right)\left\|\frac{\alpha_t e_k - x}{\sigma_t^2} - \bar{g}\right\|^2. \tag{57}$$

$\square$

This formulation reveals that score variance is low when the posterior is sharply peaked (i.e., $x$ is close to a single one-hot vector), and increases when the posterior mass is spread over multiple categories. The minimal $n$-singular point case is recovered when $\pi_k = \frac{1}{n}$ for $k \in \mathcal{S}$ and $x = \frac{1}{n}\sum_{k \in \mathcal{S}} e_k$, confirming the consistency of both analyses.

## B   Implementation

The following delineates the foundation of our experiments:

- Codebase: Python & PyTorch

- CPU: AMD EPYC-Rome 7002

- GPU: NVIDIA A100 80GB PCIe

### B.1   Categorical Representations

**One-Hot Encoding**. The one-hot encoding (ONEHOT) representation of $\mathbf{c}^{(i)}$ is constructed by concatenating the one-hot encoded vectors of each individual feature $c_j^{(i)}$. Specifically, the one-hot encoding for $c_j^{(i)}$ is a vector $\mathbf{e}(c_j^{(i)}) \in \{0,1\}^{K_j}$, where the $k$-th entry is defined as:

$$\mathbf{e}(c_j^{(i)})_k = \begin{cases} 1 & \text{if } k = c_j^{(i)}, \\ 0 & \text{otherwise,} \end{cases} \tag{58}$$

for $k \in \{1, 2, \ldots, K_j\}$. The one-hot encoded representation of $\mathbf{c}^{(i)}$ is then:

$$\text{ONEHOT}(\mathbf{c}^{(i)}) = [\mathbf{e}(c_1^{(i)}), \mathbf{e}(c_2^{(i)}), \ldots, \mathbf{e}(c_{D_{\text{cat}}}^{(i)})]. \tag{59}$$

The resulting vector has a total length of $\sum_{j=1}^{D_{\text{cat}}} K_j$, which corresponds to the sum of the unique categories across all categorical features. A softmax or a logarithm is then applied to the one-hot representation to yield a continuous probability distribution.

**Learned Embeddings**. Learned Embeddings (LEARNED) (Mikolov et al., 2013) encode the categorical component $\mathbf{c}^{(i)}$ using a representation trained directly within the model. Specifically, each categorical feature $c_j^{(i)}$ (the $j$-th element of $\mathbf{c}^{(i)}$) is assigned a trainable embedding vector of fixed dimensionality $d_{\text{EMB}}$. Hence, for each $j \in \{1, \ldots, D_{\text{cat}}\}$, we have an embedding matrix

$$E_j \in \mathbb{R}^{K_j \times d_{\text{EMB}}}.$$

The embedding lookup operation retrieves the embedding vector for each $c_j^{(i)}$, and these vectors are then concatenated to form the full embedding for the categorical features:

$$\text{LEARNED}(\mathbf{c}^{(i)}) = \text{concat}\big(E_1[c_1^{(i)}],\ E_2[c_2^{(i)}],\ \ldots,\ E_{D_{\text{cat}}}[c_{D_{\text{cat}}}^{(i)}]\big) \in \mathbb{R}^{D_{\text{cat}} \cdot d_{\text{EMB}}}. \tag{60}$$

To decode the learned embeddings back into categorical values, a nearest-neighbor approach is applied. For each embedding segment corresponding to a categorical feature, the pairwise distance between the embedding and the learned weights is computed, and the category with the minimum distance is selected:

$$\hat{c}_j^{(i)} \;=\; \underset{k \in \{1,\ldots,K_j\}}{\arg\min} \left\| \tilde{\mathbf{c}}_j^{(i)} - E_j[k] \right\|, \tag{61}$$

where $\tilde{\mathbf{c}}_j^{(i)} \in \mathbb{R}^{d_{\text{EMB}}}$ is the segment of the concatenated embedding corresponding to the $j$-th categorical feature, and $E_j \in \mathbb{R}^{K_j \times d_{\text{EMB}}}$ is the embedding matrix for that feature.

**Analog Bits**. Analog Bits (I2B) (Chen et al., 2022) encode categorical features using a binary-based continuous representation. The encoding process involves two steps. We convert the categorical value to a real-valued binary representation where each category can be expressed using $\lceil \log_2(K) \rceil$ binary bits based on the number of categories:

$$\text{I2B}(c_j^{(i)}, K_j^{(i)}) \in \mathbb{R}^{\lceil \log_2 K_j^{(i)} \rceil} \tag{62}$$

followed by a shift and scale formula:

$$\text{I2B}(c_j^{(i)}, K_j^{(i)}) \leftarrow \left( \text{I2B}(c_j^{(i)}, K_j^{(i)}) \cdot 2 - 1 \right). \tag{63}$$

Thus, training and sampling of continuous-feature generative models (e.g., diffusion models) become computationally tractable. To decode, thresholding and rounding are applied to the generated continuous bits from the model to convert them back into binary form, which can be decoded trivially back into the original categorical values.

**Dictionary**. Dictionary encoding (DIC) represents categorical features using a continuous look-up embedding function. The encoding assigns equally spaced real-valued representations within a tunable specified range, $[-1, 1]$, to balance sparsity and separability. For categorical features with more categories, a wider range may be necessary to ensure proper distinction between values. The encoding process is defined as:

$$\text{DIC}(c_j^{(i)}, K_j^{(i)}) = -1 + \frac{2c_j^{(i)}}{K_j^{(i)} - 1}, \quad \text{DIC}(c_j^{(i)}, K_j^{(i)}) \in [-1, 1] \tag{64}$$

To decode, the nearest embedding is determined by comparing the encoded continuous value with all $K_j^{(i)}$ possible embeddings, selecting the category with the smallest Euclidean distance.

## B.2 Architecture

**Out-of-index (OOI)**. Out-of-index (OOI) can potentially occur due to the generative nature of DDPM and FM. For CatConverter, OOI values are cast to the value of the 0-th index. Although casting ensures that all generated categorical values fall within the valid range, it introduces a bias. It can slightly inflate the frequency of the 0-th category, particularly for features with class imbalance. This effect could, in principle, skew the marginal distribution of certain categorical variables. We conducted additional experiments highlighting the casting rate that occurs when using our method. As observed in Table 8, casting rate is relatively low for most datasets, ranging between 5% to 20% apart from the Stroke dataset at around 30%. Given the low casting rates observed, the overall impact of this bias appears minimal in practice. Nonetheless, we still achieve exceptional results across all datasets and benchmarks, validating the effectiveness of our method.

Table 8: Out-of-Index Casting Rate

| METHODS | ADULT | DEFAULT | SHOPPERS | STROKE | DIABETES | BEIJING | NEWS |
|---|---|---|---|---|---|---|---|
| TABREP-DDPM | $0.100_{\pm.001}$ | $0.094_{\pm.001}$ | $0.112_{\pm.005}$ | $0.272_{\pm.003}$ | $0.183_{\pm.002}$ | $0.072_{\pm.002}$ | $0.092_{\pm.003}$ |
| TABREP-FLOW | $0.108_{\pm.001}$ | $0.069_{\pm.001}$ | $0.138_{\pm.001}$ | $0.348_{\pm.002}$ | $0.149_{\pm.001}$ | $0.054_{\pm.001}$ | $0.093_{\pm.001}$ |

**Flow Matching/DDPM Denoising Network**. The input layer projects the batch of tabular data input samples $\mathbf{z}_t$, each with dimension $d_{in}$, to the dimensionality $d_t$ of our time step embeddings $t_{emb}$ through a fully connected layer. This is so that we may leverage temporal information, which is appended to the result of the projection in the form of sinusoidal time step embeddings.

$$h_{in} = \texttt{FC}_{d_t}(\mathbf{z}_t) + t_{emb} \tag{65}$$

Subsequently, the output is passed through hidden layers $h_1$, $h_2$, $h_3$, and $h_4$ which are fully connected networks used to learn the denoising direction or vector field. The output dimension of each layer is chosen as $d_t$, $2d_t$, $2d_t$, and $d_t$ respectively. On top of the FC networks, each layer also consists of an activation function followed by dropout, as seen in the formulas below. This formulation is repeated for each hidden layer, at the end of which we obtain $h_{out}$. The exact activations, dropout, and other hyperparameters chosen are shown in Table 9.

$$h_1 = \texttt{Dropout}(\texttt{Activation}(\texttt{FC}(h_{in}))) \tag{66}$$

At last, the output layer transforms $h_{out}$, of dimension $t_{emb}$ back to dimension $d_{in}$ through a fully connected network, which now represents the score function $\nabla_{\mathbf{z}_t} \log p_\theta(\mathbf{z}_t)$.

$$\nabla_{\mathbf{z}_t} \log p_\theta(\mathbf{z}_t) = \texttt{FC}_{d_{in}}(h_{out}) \tag{67}$$

**QuantileTransformer** (Zhang et al., 2023; Pedregosa et al., 2011). To normalize continuous features, we adopt a quantile-based transformation that maps each feature's empirical cumulative distribution function (CDF) to a target normal distribution. Each value is replaced by its corresponding quantile rank within the training data and subsequently projected through the inverse CDF of a Gaussian distribution. This procedure produces approximately Gaussian marginals, ensuring that continuous variables are smoothly distributed and well-scaled, which in turn stabilizes the training dynamics of diffusion-based models. The number of quantiles is determined adaptively as $\max(\min(N/30, 1000), 10)$, where $N$ denotes the number of training samples.

### B.3 Hyperparameters

The hyperparameters selected for our model are shown in Table 9. The remaining hyperparameters of the baselines are tuned accordingly per its paper's instructions. In terms of hyperparameter sensitivity of TabRep, we find that the learning rate, weight decay, and batch size have minimal impact on the downstream performance. The impact of training iterations and sampling steps can be found in Table 5.

Table 9: TABREP Hyperparameters.

| General | | Flow Matching/DDPM Denoising MLP | |
|---|---|---|---|
| **Hyperparameter** | **Value** | **Hyperparameter** | **Value** |
| Training Iterations | $100,000$ | Timestep embedding dimension $d_t$ | $1024$ |
| Flow Matching/DDPM Sampling Steps | $50/1000$ | Activation | ReLU |
| Learning Rate | $1e{-}4$ | Dropout | $0.0$ |
| Weight Decay | $5e{-}4$ | Hidden layer dimension $[h1, h2, h3, h4]$ | $[1024, 2048, 2048, 1024]$ |
| Batch Size | $4096$ | | |
| Optimizer | Adam | | |

## C Experiments

### C.1 Datasets

Experiments were conducted with a total of 7 tabular datasets from the UCI Machine Learning Repository (Dua & Graff, 2017) with a (CC-BY 4.0) license. Classification tasks were performed on the Adult, Default, Shoppers, Stroke, and Diabetes datasets, while regression tasks were performed on the Beijing and News datasets. Each dataset was split into training, validation, and testing sets with a ratio of 8:1:1, except for the

Adult dataset, whose official testing set was used and the remainder split into training and validation sets with an 8:1 ratio, and the Diabetes dataset, which was split into a ratio of 6:2:2. The resulting statistics of each dataset are shown in Table 10.

Table 10: Dataset Statistics. "# Num" and "# Cat" refer to the number of numerical and categorical columns.

| Dataset | # Samples | # Num | # Cat | # Max Cat | # Train | # Validation | # Test | Task Type |
|---------|-----------|-------|-------|-----------|---------|--------------|--------|-----------|
| **Adult** | $48,842$ | 6 | 9 | 42 | $28,943$ | $3,618$ | $16,281$ | Binary Classification |
| **Default** | $30,000$ | 14 | 11 | 11 | $24,000$ | $3,000$ | $3,000$ | Binary Classification |
| **Shoppers** | $12,330$ | 10 | 8 | 20 | $9,864$ | $1,233$ | $1,233$ | Binary Classification |
| **Beijing** | $41,757$ | 7 | 5 | 31 | $33,405$ | $4,175$ | $4,175$ | Regression |
| **News** | $39,644$ | 46 | 2 | 7 | $31,714$ | $3,965$ | $3,965$ | Regression |
| **Stroke** | $4,909$ | 3 | 8 | 5 | $3,927$ | 490 | 490 | Binary Classification |
| **Diabetes** | $99,473$ | 8 | 21 | 10 | $59,683$ | $19,895$ | $19,895$ | Multiclass Classification |

## C.2 Baselines

TabRep's performance is evaluated in comparison to previous works in diffusion-based mixed-type tabular data generation. This includes STaSy (Kim et al., 2022), CoDi (Lee et al., 2023), TabDDPM (Kotelnikov et al., 2023), TabSYN (Zhang et al., 2023), and TabDiff (Shi et al., 2024b). The underlying architectures and implementation details of these models are presented in Table 11. Note that different benchmarks were used for CDTD (Mueller et al., 2025) thus, we decided to not include it in our baselines.

## C.3 Benchmarks

We evaluate the generative performance on a broad suite of benchmarks. We analyze the capabilities in *downstream tasks* such as machine learning efficiency (MLE), where we determine the AUC score for classification tasks and RMSE for regression tasks of XGBoost (Chen & Guestrin, 2016) on the generated synthetic datasets. Next, we conduct experiments on *low-order statistics* where we perform column-wise density estimation (CDE) and pair-wise column correlation (PCC). Lastly, we examine the models' quality on *high-order metrics* such as $\alpha$-*precision* and $\beta$-*recall* scores (Alaa et al., 2022). We add three additional benchmarks including a detection test metric, Classifier Two Sample Tests (C2ST) (SDMetrics, 2024), and privacy preservation metrics: the precision and recall of a membership inference attack (MIA) (Shokri et al., 2017). In this section, we expand on the concrete formulations behind our benchmarks including machine learning efficiency, low-order statistics, and high-order metrics. We also provide an overview on the detection and privacy metrics used in our experiments.

**Machine Learning Efficiency**. To evaluate the quality of our generated synthetic data, we use the data to train a classification/regression model, using XGBoost (Chen & Guestrin, 2016). This model is applied to the real test set. *AUC* (Area Under Curve) is used to evaluate the efficiency of our model in binary classification tasks. It measures the area under the Receiver Operating Characteristic (or ROC) curve, which plots the True Positive Rate against the False Positive Rate. AUC may take values in the range [0,1]. A higher AUC value suggests that our model achieves a better performance in binary classification tasks and vice versa.

$$\text{AUC} = \int_0^1 \text{TPR}(\text{FPR}) \, d(\text{FPR}) \tag{68}$$

*RMSE* (Root Mean Square Error) is used to evaluate the efficiency of our model in regression tasks. It measures the average magnitude of the deviations between predicted values ($\hat{y}_i$) and actual values ($y_i$). A smaller RMSE model indicates a better fit of the model to the data.

$$\text{RMSE} = \sqrt{\frac{1}{n} \sum_{i=1}^{n} (y_i - \hat{y}_i)^2} \tag{69}$$

Table 11: Comparison of tabular data synthesis baselines.

| Method | Model[1] | Type[2] | Categorical Encoding | Numerical Encoding | Additional Techniques |
|---|---|---|---|---|---|
| **STaSy** | Score-based Diffusion | U | One-Hot Encoding | Min-max scaler | Self-paced learning and fine-tuning. |
| **CoDi** | DDPM/ Multino-mial Diffusion | S | One-Hot Encoding | Min-max scaler | Model Inter-conditioning and Contrastive learning to learn dependencies between categorical and numerical data. |
| **TabDDPM** | DDPM/ Multino-mial Diffusion | S | One-Hot Encoding | Quantile Transformer | Concatenation of numerical and categorical features. |
| **TabSYN** | VAE + EDM | U | VAE-Learned | Quantile Transformer | Feature Tokenizer and Transformer encoder to learn cross-feature relationships with adaptive loss weighing to increase reconstruction performance. |
| **TabDiff** | EDM/Masked Diffusion | S | One-Hot Encoding | Quantile Transformer | Joint continuous-time diffusion process of numerical and categorical variables under learnable noise schedules, with a stochastic sampler to correct sampling errors. |
| **TabRep-DDPM** | DDPM | U | CAT-CONVERTER | Quantile Transformer | Plug-and-play for Diffusion Models. |
| **TabRep-Flow** | Flow Matching | U | CAT-CONVERTER | Quantile Transformer | Plug-and-play for Diffusion Models. |

[1] The "Model" Column indicates the underlying architecture used for the model. Options include Denoising Diffusion Probabilistic Models or DDPMs (Ho et al., 2020b), Multinomial Diffusion (Hoogeboom et al., 2021), EDM, as introduced in (Karras et al., 2022).

[2] The "Type" column indicates the data integration approach used in the model. "U" denotes a unified data space where numerical and categorical data are combined after initial processing and fed collectively into the model. "S" represents a separated data space, where numerical and categorical data are processed and fed into distinct models.

**Low-Order Statistics**. *Column-wise Density Estimation* between numerical features is achieved with the Kolmogorov-Smirnov Test (KST). The Kolmogorov-Smirnov statistic is used to evaluate how much two underlying one-dimensional probability distributions differ, and is characterized by the below equation:

$$\text{KST} = \sup_x |F_1(x) - F_2(x)|, \tag{70}$$

where $F_n(x)$, the empirical distribution function of sample n is calculated by

$$\text{F}_\text{n}(\text{x}) = \frac{1}{n} \sum_{i=1}^{n} \mathbf{1}_{(-\infty, x]}(X_i) \tag{71}$$

*Column-wise Density Estimation* between two categorical features is determined by calculating the Total Variation Distance (TVD). This statistic captures the largest possible difference in the probability of any

event under two different probability distributions. It is expressed as

$$\text{TVD} = \frac{1}{2} \sum_{x \in X} |P_1(x) - P_2(x)|, \tag{72}$$

where $P_1(x)$ and $P_2(x)$ are the probabilities (PMF) assigned to data point x by the two sample distributions respectively.

*Pair-wise Column Correlation* between two numerical features is computed using the Pearson Correlation Coefficient (PCC). It assigns a numerical value to represent the linear relationship between two columns, ranging from -1 (perfect negative linear correlation) to +1 (perfect positive linear correlation), with 0 indicating no linear correlation. It is computed as:

$$\rho(x, y) = \frac{\text{cov}(x, y)}{\sigma_x \sigma_y}, \tag{73}$$

To compare the Pearson Coefficients of our real and synthetic datasets, we quantify the dissimilarity in pair-wise column correlation between two samples

$$\text{Pearson Score} = \frac{1}{2} \mathbb{E}_{x,y} |\rho^1(x, y) - \rho^2(x, y)| \tag{74}$$

*Pair-wise Column Correlation* between two categorical features in a sample is characterized by a Contingency Table. This table is constructed by tabulating the frequencies at which specific combinations of the levels of two categorical variables work and recording them in a matrix format.

To quantify the dissimilarity of contingency matrices between two different samples, we use the Contingency Score.

$$\text{Contingency Score} = \frac{1}{2} \sum_{\alpha \in A} \sum_{\beta \in B} |P_{1,(\alpha,\beta)} - P_{2,(\alpha,\beta)}|, \tag{75}$$

where $\alpha$ and $\beta$ describe possible categorical values that can be taken in features $A$ and $B$. $P_{1,(\alpha,\beta)}$ and $P_{2,(\alpha,\beta)}$ refer to the contingency tables representing the features $\alpha$ and $\beta$ in our two samples, which in this case corresponds to the real and synthetic datasets.

In order to obtain the column-wise density estimation and pair-wise correlation between a categorical and a numerical feature, we bin the numerical data into discrete categories before applying TVD and Contingency score respectively to obtain our low-order statistics.

We utilize the implementation of these experiments as provided by the SDMetrics library[1].

**High-Order Statistics**. We utilize the implementations of High-Order Statistics as provided by the synthcity[2] library. *α-precision* measures the overall fidelity of the generated data and is an extension of the classical machine learning quality metric of "precision". This formulation is based on the assumption that $\alpha$ fraction of our real samples are characteristic of the original data distribution and the rest are outliers. α-precision therefore quantifies the percentage of generated synthetic samples that match $\alpha$ fraction of real samples. *β-recall* characterizes the diversity of our synthetic data and is similarly based on the quality metric of "recall". β-recall shares a similar assumption as α-precision, except that we now assume that $\beta$ fraction of our synthetic samples are characteristic of the distribution. Therefore, this measure obtains the fraction of the original data distribution represented by the $\beta$ fraction of our generated samples (Alaa et al., 2022).

**Detection Metric: Classifier Two-Sample Test (C2ST)**. The Classifier Two-Sample Test, a detection metric, assesses the ability to distinguish real data from synthetic data. This is done through a machine learning model that attempts to label whether a data point is synthetic or real. The score ranges from 0 to 1 where a score closer to 1 is superior, indicating that the machine learning model cannot concretely identify whether the data point is real or generated. We select logistic regression as our machine learning model, using the implementation provided by SDMetric (SDMetrics, 2024).

---

[1]https://github.com/sdv-dev/SDMetrics
[2]https://github.com/vanderschaarlab/synthcity

**Privacy Metric: Membership Inference Attacks (MIA)**. Membership inference attacks evaluate the vulnerability of machine learning models to privacy leakage by determining whether a given instance was included in the training dataset (Shokri et al., 2017). The attacker often constructs a shadow model to mimic the target model's behavior and trains a binary classifier to distinguish membership status based on observed patterns. We replaced DCR with Membership Inference Attacks since existing privacy ML literature (Ganev & Cristofaro, 2024; Ward et al., 2024) conducted research highlighting the "Inadequacy of Similarity-based Privacy Metrics" such as DCR.

These attacks are evaluated using precision, the fraction of inferred members that are true members, and recall, the fraction of true members correctly identified. When records are equally balanced between members and non-members, the ideal precision and recall are 0.5, indicating that the attack is no better than random guessing. Higher values suggest privacy leakage and reveal vulnerabilities in the model. Implementation of this metric is provided by SynthEval (Lautrup et al., 2024).

# D  Further Experimental Results

We perform experiments using several diffusion-based tabular generative model baselines, including STaSy (Kim et al., 2022), CoDi (Lee et al., 2023), TabDDPM (Kotelnikov et al., 2023), TabSYN (Zhang et al., 2023). We include TabDiff's reported results as the authors have not released code (Shi et al., 2024b).

We also incorporate several ablation experiments. We introduce TabFlow, an adaptation of TabDDPM (Kotelnikov et al., 2023) that models numerical and categorical tabular data with continuous flow matching (Lipman et al., 2022; Liu et al., 2022) and discrete flow matching (Tong et al., 2023), to examine the effect of unifying the data space. Experiments are also performed on various categorical representations on both diffusion and flow models. This includes one-hot, 1D-Learned Embedding, 2D-Learned Embedding, and Analog Bits (Chen et al., 2022), to demonstrate TABREP's effectiveness. We show that our proposed TABREP framework achieves superior performance on the vast majority of metrics in Appendix D.1.

We evaluate AUC (classification), RMSE (regression), Column-Wise Density Estimation (CDE), Pair-Wise Column Correlation (PCC), $\alpha$-Precision, $\beta$-Recall scores, Classifier-Two Sample Test scores (C2ST), and Membership Inference Attacks Precision (MIA P.) and Recall (MIA R.) scores for our 7 datasets. $\uparrow$ indicates that the higher the score, the better the performance; $\downarrow$ indicates that the lower the score, the better the performance; $\updownarrow$ indicates that an optimal score should be as close to 50% as possible.

The metrics and error bars shown in the tables in this section are derived from the mean and standard deviation of the experiments performed on 20 sampling iterations on the best-validated model.

## D.1 Additional Baseline Results

| | Adult | | | | | | | |
|---|---|---|---|---|---|---|---|---|
| METHODS | AUC ↑ | CDE ↑ | PCC ↑ | $\alpha$ ↑ | $\beta$ ↑ | C2ST ↑ | MIA P. ↕ | MIA R. ↕ |
| STASY | $0.906_{\pm.001}$ | $92.26_{\pm0.04}$ | $89.15_{\pm0.10}$ | $77.05_{\pm0.29}$ | $33.54_{\pm0.36}$ | $55.37$ | $49.52_{\pm0.50}$ | $24.51_{\pm0.44}$ |
| CODI | $0.871_{\pm0.006}$ | $74.28_{\pm0.08}$ | $77.38_{\pm0.21}$ | $74.45_{\pm0.35}$ | $8.74_{\pm0.16}$ | $15.80$ | $55.00_{\pm7.26}$ | $0.05_{\pm0.01}$ |
| TABDDPM | $0.910_{\pm0.001}$ | $98.37_{\pm0.08}$ | $96.69_{\pm0.09}$ | $90.99_{\pm0.35}$ | $\mathbf{62.19_{\pm0.48}}$ | $97.55$ | $51.30_{\pm0.07}$ | $56.90_{\pm0.29}$ |
| TABSYN | $0.906_{\pm0.001}$ | $98.89_{\pm0.03}$ | $97.56_{\pm0.06}$ | $98.97_{\pm0.26}$ | $47.68_{\pm0.27}$ | $95.49$ | $50.91_{\pm0.16}$ | $42.91_{\pm0.31}$ |
| TABDIFF | $0.912_{\pm0.002}$ | $99.37_{\pm0.05}$ | $98.51_{\pm0.16}$ | $99.02_{\pm0.20}$ | $51.64_{\pm0.20}$ | $\mathbf{99.50}$ | $51.03_{\pm0.12}$ | $52.00_{\pm0.35}$ |
| TABREP-DDPM | $\mathbf{0.913_{\pm0.002}}$ | $\mathbf{99.39_{\pm0.04}}$ | $\mathbf{98.63_{\pm0.04}}$ | $\mathbf{99.11_{\pm0.25}}$ | $52.04_{\pm0.12}$ | $\mathbf{99.50}$ | $\mathbf{50.44_{\pm0.83}}$ | $52.78_{\pm0.21}$ |
| TABREP-FLOW | $0.912_{\pm0.002}$ | $98.63_{\pm0.02}$ | $97.55_{\pm0.23}$ | $98.21_{\pm0.34}$ | $49.91_{\pm0.28}$ | $95.48$ | $50.65_{\pm0.20}$ | $\mathbf{50.51_{\pm0.21}}$ |

| | Default | | | | | | | |
|---|---|---|---|---|---|---|---|---|
| METHODS | AUC ↑ | CDE ↑ | PCC ↑ | $\alpha$ ↑ | $\beta$ ↑ | C2ST ↑ | MIA P. ↕ | MIA R. ↕ |
| STASY | $0.752_{\pm0.006}$ | $89.41_{\pm0.03}$ | $92.64_{\pm0.03}$ | $94.83_{\pm0.15}$ | $40.23_{\pm0.22}$ | $62.82$ | $51.84_{\pm0.69}$ | $30.37_{\pm0.99}$ |
| CODI | $0.525_{\pm0.006}$ | $81.07_{\pm0.08}$ | $86.25_{\pm0.73}$ | $81.21_{\pm0.11}$ | $19.75_{\pm0.30}$ | $42.28$ | $46.66_{\pm2.54}$ | $3.41_{\pm0.53}$ |
| TABDDPM | $0.761_{\pm0.004}$ | $98.20_{\pm0.05}$ | $97.16_{\pm0.19}$ | $96.78_{\pm0.30}$ | $\mathbf{53.73_{\pm0.28}}$ | $97.12$ | $51.34_{\pm0.49}$ | $44.96_{\pm0.59}$ |
| TABSYN | $0.755_{\pm0.005}$ | $98.61_{\pm0.08}$ | $\mathbf{98.33_{\pm0.67}}$ | $98.49_{\pm0.20}$ | $46.06_{\pm0.37}$ | $95.83$ | $50.80_{\pm0.56}$ | $43.71_{\pm0.89}$ |
| TABDIFF | $0.763_{\pm0.005}$ | $98.76_{\pm0.07}$ | $97.45_{\pm0.75}$ | $98.49_{\pm0.28}$ | $51.09_{\pm0.25}$ | $97.74$ | $51.15_{\pm0.62}$ | $46.67_{\pm0.55}$ |
| TABREP-DDPM | $0.764_{\pm0.005}$ | $\mathbf{98.97_{\pm0.19}}$ | $96.74_{\pm0.62}$ | $\mathbf{98.66_{\pm0.24}}$ | $48.22_{\pm0.48}$ | $\mathbf{98.90}$ | $50.07_{\pm0.41}$ | $\mathbf{48.96_{\pm0.41}}$ |
| TABREP-FLOW | $\mathbf{0.782_{\pm0.005}}$ | $97.45_{\pm0.06}$ | $92.86_{\pm1.75}$ | $96.50_{\pm0.44}$ | $49.99_{\pm0.23}$ | $89.36$ | $\mathbf{50.04_{\pm0.95}}$ | $47.07_{\pm0.40}$ |

| | Shoppers | | | | | | | |
|---|---|---|---|---|---|---|---|---|
| METHODS | AUC ↑ | CDE ↑ | PCC ↑ | $\alpha$ ↑ | $\beta$ ↑ | C2ST ↑ | MIA P. ↕ | MIA R. ↕ |
| STASY | $0.914_{\pm0.005}$ | $82.53_{\pm0.19}$ | $81.40_{\pm0.27}$ | $68.18_{\pm0.29}$ | $26.24_{\pm0.60}$ | $25.82$ | $51.23_{\pm2.38}$ | $17.54_{\pm0.19}$ |
| CODI | $0.865_{\pm0.006}$ | $67.27_{\pm0.03}$ | $80.52_{\pm0.12}$ | $90.52_{\pm0.37}$ | $19.22_{\pm0.12}$ | $19.04$ | $59.67_{\pm11.74}$ | $1.36_{\pm0.45}$ |
| TABDDPM | $0.915_{\pm0.004}$ | $97.58_{\pm0.18}$ | $96.72_{\pm0.22}$ | $90.85_{\pm0.60}$ | $72.46_{\pm0.46}$ | $83.49$ | $51.24_{\pm1.48}$ | $46.08_{\pm1.57}$ |
| TABSYN | $0.918_{\pm0.004}$ | $96.00_{\pm0.12}$ | $95.18_{\pm0.11}$ | $96.28_{\pm0.24}$ | $45.79_{\pm0.31}$ | $83.77$ | $51.00_{\pm1.08}$ | $42.14_{\pm0.76}$ |
| TABDIFF | $0.919_{\pm0.005}$ | $98.72_{\pm0.09}$ | $\mathbf{98.26_{\pm0.08}}$ | $\mathbf{99.11_{\pm0.34}}$ | $49.75_{\pm0.64}$ | $\mathbf{98.43}$ | $51.11_{\pm1.11}$ | $46.86_{\pm1.20}$ |
| TABREP-DDPM | $\mathbf{0.926_{\pm0.005}}$ | $\mathbf{98.97_{\pm0.10}}$ | $97.62_{\pm0.02}$ | $96.14_{\pm0.19}$ | $53.68_{\pm0.73}$ | $96.37$ | $\mathbf{49.86_{\pm0.98}}$ | $48.16_{\pm0.90}$ |
| TABREP-FLOW | $0.919_{\pm0.005}$ | $97.74_{\pm0.03}$ | $97.08_{\pm0.07}$ | $95.85_{\pm0.46}$ | $\mathbf{55.92_{\pm0.37}}$ | $94.20$ | $51.38_{\pm1.66}$ | $\mathbf{49.19_{\pm0.86}}$ |

| | Stroke | | | | | | | |
|---|---|---|---|---|---|---|---|---|
| METHODS | AUC ↑ | CDE ↑ | PCC ↑ | $\alpha$ ↑ | $\beta$ ↑ | C2ST ↑ | MIA P. ↕ | MIA R. ↕ |
| STASY | $0.833_{\pm0.03}$ | $89.36_{\pm0.13}$ | $84.99_{\pm0.09}$ | $91.49_{\pm0.33}$ | $39.92_{\pm0.76}$ | $40.64$ | $54.22_{\pm1.14}$ | $34.63_{\pm1.39}$ |
| CODI | $0.798_{\pm0.032}$ | $87.42_{\pm0.17}$ | $80.65_{\pm1.81}$ | $86.46_{\pm0.53}$ | $28.59_{\pm0.47}$ | $24.47$ | $54.54_{\pm1.60}$ | $27.32_{\pm3.50}$ |
| TABDDPM | $0.808_{\pm0.033}$ | $99.10_{\pm0.05}$ | $97.09_{\pm1.17}$ | $98.05_{\pm0.14}$ | $\mathbf{71.42_{\pm0.30}}$ | $100.00$ | $53.45_{\pm2.89}$ | $55.12_{\pm0.81}$ |
| TABSYN | $0.845_{\pm0.035}$ | $96.79_{\pm0.08}$ | $95.18_{\pm0.22}$ | $95.49_{\pm0.41}$ | $48.85_{\pm0.26}$ | $89.93$ | $51.11_{\pm1.54}$ | $47.97_{\pm1.27}$ |
| TABDIFF | $0.848_{\pm0.021}$ | $99.09_{\pm0.12}$ | $\mathbf{97.91_{\pm0.22}}$ | $\mathbf{98.95_{\pm0.53}}$ | $49.91_{\pm0.86}$ | $99.87$ | $52.20_{\pm1.61}$ | $47.15_{\pm1.30}$ |
| TABREP-DDPM | $\mathbf{0.869_{\pm0.027}}$ | $\mathbf{99.14_{\pm0.20}}$ | $97.11_{\pm0.60}$ | $98.32_{\pm0.82}$ | $57.17_{\pm0.77}$ | $100.00$ | $51.74_{\pm1.85}$ | $\mathbf{50.89_{\pm0.93}}$ |
| TABREP-FLOW | $0.854_{\pm0.028}$ | $98.42_{\pm0.31}$ | $97.37_{\pm2.12}$ | $96.40_{\pm0.71}$ | $63.91_{\pm0.87}$ | $95.96$ | $\mathbf{50.66_{\pm1.77}}$ | $49.43_{\pm1.16}$ |

| | Diabetes | | | | | | | |
|---|---|---|---|---|---|---|---|---|
| METHODS | F1 ↑ | CDE ↑ | PCC ↑ | $\alpha$ ↑ | $\beta$ ↑ | C2ST ↑ | MIA P. ↕ | MIA R. ↕ |
| STASY | $0.374_{\pm0.003}$ | $95.25_{\pm0.03}$ | $93.41_{\pm0.08}$ | $90.00_{\pm0.17}$ | $39.62_{\pm0.23}$ | $54.71$ | $49.47_{\pm0.22}$ | $29.75_{\pm0.16}$ |
| CODI | $0.288_{\pm0.009}$ | $76.42_{\pm0.02}$ | $78.07_{\pm0.18}$ | $38.96_{\pm0.16}$ | $6.39_{\pm0.16}$ | $3.95$ | $0.00_{\pm0.00}$ | $0.00_{\pm0.00}$ |
| TABDDPM | $0.376_{\pm0.003}$ | $99.26_{\pm0.01}$ | $98.71_{\pm0.01}$ | $95.36_{\pm0.32}$ | $\mathbf{52.65_{\pm0.03}}$ | $92.18$ | $50.40_{\pm0.65}$ | $52.06_{\pm0.11}$ |
| TABSYN | $0.361_{\pm0.001}$ | $99.04_{\pm0.01}$ | $98.32_{\pm0.03}$ | $98.08_{\pm0.14}$ | $45.08_{\pm0.04}$ | $\mathbf{93.38}$ | $50.10_{\pm0.34}$ | $44.42_{\pm0.28}$ |
| TABDIFF | $0.353_{\pm0.006}$ | $98.72_{\pm0.02}$ | $97.80_{\pm0.03}$ | $96.84_{\pm0.14}$ | $36.96_{\pm0.18}$ | $91.60$ | $49.27_{\pm0.41}$ | $31.01_{\pm0.22}$ |
| TABREP-DDPM | $0.373_{\pm0.003}$ | $\mathbf{99.36_{\pm0.02}}$ | $\mathbf{98.75_{\pm0.03}}$ | $97.19_{\pm0.22}$ | $45.75_{\pm0.49}$ | $92.65$ | $\mathbf{50.07_{\pm0.37}}$ | $\mathbf{51.43_{\pm0.13}}$ |
| TABREP-FLOW | $\mathbf{0.377_{\pm0.002}}$ | $99.00_{\pm0.02}$ | $98.46_{\pm0.05}$ | $\mathbf{99.08_{\pm0.13}}$ | $48.58_{\pm0.17}$ | $90.41$ | $50.24_{\pm0.37}$ | $50.33_{\pm0.13}$ |

| | Beijing | | | | | | | |
|---|---|---|---|---|---|---|---|---|
| METHODS | RMSE ↓ | CDE ↑ | PCC ↑ | $\alpha$ ↑ | $\beta$ ↑ | C2ST ↑ | MIA P. ↕ | MIA R. ↕ |
| STASY | $0.656_{\pm0.014}$ | $93.14_{\pm0.07}$ | $90.63_{\pm0.11}$ | $96.41_{\pm0.10}$ | $51.35_{\pm0.16}$ | $77.80$ | $48.98_{\pm0.78}$ | $34.06_{\pm0.33}$ |
| CODI | $0.818_{\pm0.021}$ | $83.54_{\pm0.04}$ | $90.35_{\pm0.21}$ | $96.89_{\pm0.14}$ | $53.16_{\pm0.12}$ | $80.27$ | $39.19_{\pm6.35}$ | $0.40_{\pm0.09}$ |
| TABDDPM | $0.592_{\pm0.012}$ | $99.09_{\pm0.02}$ | $96.71_{\pm0.18}$ | $97.74_{\pm0.06}$ | $\mathbf{73.13_{\pm0.26}}$ | $95.13$ | $50.43_{\pm0.58}$ | $48.35_{\pm0.79}$ |
| TABSYN | $0.555_{\pm0.013}$ | $98.34_{\pm0.01}$ | $96.85_{\pm0.24}$ | $98.08_{\pm0.27}$ | $55.68_{\pm0.16}$ | $92.92$ | $51.31_{\pm0.42}$ | $46.53_{\pm0.52}$ |
| TABDIFF | $0.555_{\pm0.013}$ | $98.97_{\pm0.05}$ | $\mathbf{97.41_{\pm0.15}}$ | $98.06_{\pm0.24}$ | $59.63_{\pm0.23}$ | $97.81$ | $50.39_{\pm0.46}$ | $48.20_{\pm0.65}$ |
| TABREP-DDPM | $\mathbf{0.508_{\pm0.006}}$ | $\mathbf{99.11_{\pm0.03}}$ | $96.97_{\pm0.20}$ | $\mathbf{98.98_{\pm0.16}}$ | $64.08_{\pm0.18}$ | $\mathbf{98.16}$ | $51.02_{\pm0.42}$ | $\mathbf{49.50_{\pm0.52}}$ |
| TABREP-FLOW | $0.536_{\pm0.006}$ | $98.28_{\pm0.07}$ | $96.92_{\pm0.21}$ | $98.16_{\pm0.13}$ | $62.65_{\pm0.12}$ | $92.26$ | $\mathbf{50.35_{\pm0.60}}$ | $49.14_{\pm0.81}$ |

| | News | | | | | | | |
|---|---|---|---|---|---|---|---|---|
| METHODS | RMSE ↓ | CDE ↑ | PCC ↑ | α ↑ | β ↑ | C2ST ↑ | MIA P. ↕ | MIA R.↕ |
| STASY | $0.871_{\pm0.002}$ | $90.50_{\pm0.10}$ | $96.59_{\pm0.02}$ | $\mathbf{97.95_{\pm0.14}}$ | $38.68_{\pm0.30}$ | 51.72 | $49.68_{\pm0.70}$ | $23.49_{\pm0.67}$ |
| CODI | $1.21_{\pm0.005}$ | $70.82_{\pm0.01}$ | $95.44_{\pm0.05}$ | $86.03_{\pm0.12}$ | $35.01_{\pm0.13}$ | 9.35 | $40.00_{\pm24.49}$ | $0.04_{\pm0.02}$ |
| TABDDPM | $3.46_{\pm1.25}$ | $94.79_{\pm0.03}$ | $89.52_{\pm0.10}$ | $90.94_{\pm0.31}$ | $40.82_{\pm0.42}$ | 0.02 | $50.72_{\pm0.97}$ | $9.88_{\pm0.52}$ |
| TABSYN | $0.866_{\pm0.021}$ | $98.22_{\pm0.04}$ | $98.53_{\pm0.02}$ | $95.83_{\pm0.33}$ | $43.97_{\pm0.27}$ | 95.85 | $49.86_{\pm0.75}$ | $34.42_{\pm0.90}$ |
| TABDIFF | $0.866_{\pm0.021}$ | $97.65_{\pm0.03}$ | $98.72_{\pm0.04}$ | $97.36_{\pm0.17}$ | $42.10_{\pm0.32}$ | 93.08 | $52.46_{\pm0.75}$ | $16.03_{\pm0.66}$ |
| TABREP-DDPM | $0.836_{\pm0.001}$ | $\mathbf{98.46_{\pm0.01}}$ | $\mathbf{99.09_{\pm0.05}}$ | $95.35_{\pm0.11}$ | $48.49_{\pm0.12}$ | $\mathbf{96.70}$ | $49.87_{\pm0.99}$ | $\mathbf{40.10_{\pm0.55}}$ |
| TABREP-FLOW | $\mathbf{0.814_{\pm0.002}}$ | $96.89_{\pm0.03}$ | $98.34_{\pm0.29}$ | $90.91_{\pm0.25}$ | $\mathbf{51.75_{\pm0.16}}$ | 88.13 | $50.90_{\pm0.93}$ | $35.48_{\pm0.78}$ |

## D.2 Additional Ablation Results

| | Adult | | | | | | | |
|---|---|---|---|---|---|---|---|---|
| METHODS | AUC ↑ | CDE ↑ | PCC ↑ | α ↑ | β ↑ | C2ST ↑ | MIA P. ↕ | MIA R.↕ |
| ONEHOT-DDPM | $0.476_{\pm0.057}$ | $48.94_{\pm0.12}$ | $38.05_{\pm0.08}$ | $17.44_{\pm0.21}$ | $0.70_{\pm0.01}$ | 1.89 | $38.67_{\pm19.02}$ | $0.03_{\pm0.02}$ |
| LEARNED1D-DDPM | $0.611_{\pm0.008}$ | $53.44_{\pm3.39}$ | $27.97_{\pm3.71}$ | $6.64_{\pm0.05}$ | $0.00_{\pm0.01}$ | 0.00 | $10.00_{\pm10.00}$ | $0.00_{\pm0.00}$ |
| LEARNED2D-DDPM | $0.793_{\pm0.003}$ | $62.69_{\pm1.16}$ | $38.53_{\pm1.54}$ | $7.11_{\pm0.21}$ | $0.02_{\pm0.03}$ | 0.00 | $38.00_{\pm5.61}$ | $0.03_{\pm0.01}$ |
| I2B-DDPM | $0.911_{\pm0.001}$ | $99.10_{\pm0.07}$ | $97.55_{\pm0.26}$ | $98.21_{\pm0.18}$ | $47.44_{\pm0.08}$ | 98.01 | $50.56_{\pm0.15}$ | $50.68_{\pm0.87}$ |
| DIC-DDPM | $0.912_{\pm0.002}$ | $98.95_{\pm0.03}$ | $97.65_{\pm0.12}$ | $97.99_{\pm0.54}$ | $51.09_{\pm0.24}$ | 93.61 | $51.34_{\pm0.20}$ | $51.62_{\pm0.33}$ |
| ONEHOT-FLOW | $0.895_{\pm0.003}$ | $90.66_{\pm0.07}$ | $84.32_{\pm0.10}$ | $88.35_{\pm0.15}$ | $30.64_{\pm0.13}$ | 38.88 | $51.08_{\pm0.60}$ | $13.49_{\pm0.36}$ |
| LEARNED1D-FLOW | $0.260_{\pm0.014}$ | $60.00_{\pm3.13}$ | $36.20_{\pm3.61}$ | $6.82_{\pm0.08}$ | $0.02_{\pm0.03}$ | 0.00 | $24.67_{\pm10.41}$ | $0.02_{\pm0.01}$ |
| LEARNED2D-FLOW | $0.126_{\pm0.012}$ | $62.37_{\pm3.03}$ | $38.94_{\pm4.20}$ | $6.64_{\pm3.49}$ | $0.05_{\pm0.04}$ | 0.00 | $28.19_{\pm9.68}$ | $0.04_{\pm0.01}$ |
| I2B-FLOW | $0.911_{\pm0.001}$ | $98.23_{\pm0.09}$ | $97.14_{\pm0.32}$ | $\mathbf{99.54_{\pm0.31}}$ | $48.87_{\pm0.16}$ | 92.18 | $50.70_{\pm0.31}$ | $\mathbf{49.68_{\pm0.79}}$ |
| DIC-FLOW | $0.910_{\pm0.002}$ | $98.10_{\pm0.03}$ | $96.63_{\pm0.03}$ | $99.64_{\pm0.10}$ | $50.29_{\pm0.12}$ | 90.70 | $50.55_{\pm0.16}$ | $42.03_{\pm0.42}$ |
| TABFLOW | $0.908_{\pm0.002}$ | $96.21_{\pm0.05}$ | $93.59_{\pm0.05}$ | $86.76_{\pm0.28}$ | $\mathbf{53.15_{\pm0.14}}$ | 77.48 | $50.99_{\pm0.38}$ | $43.90_{\pm0.18}$ |
| TABREP-DDPM | $\mathbf{0.913_{\pm0.002}}$ | $99.39_{\pm0.04}$ | $98.63_{\pm0.04}$ | $99.11_{\pm0.25}$ | $52.04_{\pm0.12}$ | $\mathbf{99.50}$ | $50.44_{\pm0.83}$ | $52.78_{\pm0.21}$ |
| TABREP-FLOW | $0.912_{\pm0.002}$ | $98.63_{\pm0.02}$ | $97.55_{\pm0.23}$ | $98.21_{\pm0.34}$ | $49.91_{\pm0.28}$ | 95.48 | $50.65_{\pm0.20}$ | $50.51_{\pm0.21}$ |

| | Default | | | | | | | |
|---|---|---|---|---|---|---|---|---|
| METHODS | AUC ↓ | CDE ↑ | PCC ↑ | α ↑ | β ↑ | C2ST ↑ | MIA P. ↕ | MIA R.↕ |
| ONEHOT-DDPM | $0.557_{\pm0.052}$ | $50.88_{\pm0.10}$ | $50.88_{\pm0.07}$ | $4.13_{\pm0.05}$ | $0.15_{\pm0.01}$ | 0.11 | $40.00_{\pm24.49}$ | $0.08_{\pm0.05}$ |
| LEARNED1D-DDPM | $0.575_{\pm0.012}$ | $72.53_{\pm1.92}$ | $51.95_{\pm2.30}$ | $10.96_{\pm2.03}$ | $0.00_{\pm0.01}$ | 0.13 | $0.00_{\pm0.00}$ | $0.00_{\pm0.00}$ |
| LEARNED2D-DDPM | $0.290_{\pm0.009}$ | $71.11_{\pm0.32}$ | $50.79_{\pm0.36}$ | $11.99_{\pm2.54}$ | $0.01_{\pm0.01}$ | 0.06 | $15.33_{\pm11.62}$ | $0.11_{\pm0.08}$ |
| I2B-DDPM | $0.762_{\pm0.003}$ | $98.76_{\pm0.06}$ | $\mathbf{98.36_{\pm0.12}}$ | $98.20_{\pm0.17}$ | $47.46_{\pm0.37}$ | 98.34 | $50.95_{\pm0.45}$ | $45.33_{\pm0.77}$ |
| DIC-DDPM | $0.763_{\pm0.007}$ | $98.33_{\pm0.11}$ | $92.76_{\pm0.43}$ | $97.20_{\pm0.35}$ | $49.62_{\pm0.39}$ | 96.88 | $51.60_{\pm0.70}$ | $48.13_{\pm0.75}$ |
| ONEHOT-FLOW | $0.759_{\pm0.006}$ | $91.95_{\pm0.05}$ | $88.04_{\pm1.51}$ | $93.12_{\pm0.31}$ | $30.50_{\pm0.19}$ | 69.14 | $51.40_{\pm1.23}$ | $12.32_{\pm0.60}$ |
| LEARNED1D-FLOW | $0.438_{\pm0.009}$ | $73.52_{\pm0.82}$ | $53.01_{\pm0.21}$ | $20.03_{\pm5.43}$ | $0.00_{\pm0.01}$ | 0.53 | $10.00_{\pm10.00}$ | $0.05_{\pm0.05}$ |
| LEARNED2D-FLOW | $0.709_{\pm0.008}$ | $66.33_{\pm4.86}$ | $44.11_{\pm6.75}$ | $2.33_{\pm12.11}$ | $0.01_{\pm0.02}$ | 0.01 | $10.00_{\pm10.00}$ | $0.03_{\pm0.03}$ |
| I2B-FLOW | $0.763_{\pm0.004}$ | $97.67_{\pm0.13}$ | $94.65_{\pm1.28}$ | $97.42_{\pm0.57}$ | $49.15_{\pm0.48}$ | 90.04 | $50.66_{\pm0.33}$ | $43.09_{\pm1.25}$ |
| DIC-FLOW | $0.759_{\pm0.007}$ | $97.27_{\pm0.03}$ | $92.26_{\pm1.77}$ | $95.97_{\pm0.27}$ | $51.29_{\pm0.18}$ | 90.58 | $51.36_{\pm0.62}$ | $45.95_{\pm0.76}$ |
| TABFLOW | $0.742_{\pm0.005}$ | $97.38_{\pm0.03}$ | $95.01_{\pm1.44}$ | $96.92_{\pm0.12}$ | $\mathbf{53.12_{\pm0.29}}$ | 85.89 | $\mathbf{50.05_{\pm0.53}}$ | $44.11_{\pm0.65}$ |
| TABREP-DDPM | $0.764_{\pm0.005}$ | $\mathbf{98.97_{\pm0.19}}$ | $96.74_{\pm0.62}$ | $\mathbf{98.66_{\pm0.24}}$ | $48.22_{\pm0.48}$ | $\mathbf{98.90}$ | $50.07_{\pm0.41}$ | $\mathbf{48.96_{\pm0.41}}$ |
| TABREP-FLOW | $\mathbf{0.782_{\pm0.005}}$ | $97.45_{\pm0.06}$ | $92.86_{\pm1.75}$ | $96.50_{\pm0.44}$ | $49.99_{\pm0.23}$ | 89.36 | $51.02_{\pm0.95}$ | $47.07_{\pm0.40}$ |

| | Shoppers | | | | | | | |
|---|---|---|---|---|---|---|---|---|
| METHODS | AUC ↑ | CDE ↑ | PCC ↑ | α ↑ | β ↑ | C2ST ↑ | MIA P. ↕ | MIA R.↕ |
| ONEHOT-DDPM | $0.799_{\pm0.126}$ | $90.37_{\pm0.14}$ | $84.61_{\pm0.10}$ | $90.67_{\pm0.26}$ | $37.76_{\pm0.57}$ | 54.59 | $51.00_{\pm0.97}$ | $28.54_{\pm1.49}$ |
| LEARNED1D-DDPM | $0.876_{\pm0.028}$ | $78.94_{\pm4.13}$ | $62.67_{\pm6.40}$ | $21.94_{\pm7.75}$ | $1.17_{\pm0.81}$ | 1.36 | $59.09_{\pm18.85}$ | $0.84_{\pm0.26}$ |
| LEARNED2D-DDPM | $0.103_{\pm0.011}$ | $70.97_{\pm3.06}$ | $51.21_{\pm4.37}$ | $8.43_{\pm5.81}$ | $0.05_{\pm0.09}$ | 0.09 | $0.00_{\pm0.00}$ | $0.00_{\pm0.00}$ |
| I2B-DDPM | $0.919_{\pm0.005}$ | $98.67_{\pm0.05}$ | $\mathbf{98.00_{\pm0.03}}$ | $\mathbf{97.69_{\pm0.63}}$ | $50.86_{\pm0.11}$ | 96.28 | $51.77_{\pm0.98}$ | $46.80_{\pm1.91}$ |
| DIC-DDPM | $0.910_{\pm0.005}$ | $97.83_{\pm0.15}$ | $96.19_{\pm0.09}$ | $95.33_{\pm0.80}$ | $56.14_{\pm0.85}$ | 91.09 | $50.68_{\pm0.52}$ | $46.02_{\pm1.20}$ |
| ONEHOT-FLOW | $0.910_{\pm0.006}$ | $92.84_{\pm0.08}$ | $91.50_{\pm0.14}$ | $87.57_{\pm0.44}$ | $48.77_{\pm0.69}$ | 65.08 | $49.64_{\pm1.69}$ | $34.56_{\pm0.80}$ |
| LEARNED1D-FLOW | $0.134_{\pm0.015}$ | $76.25_{\pm2.16}$ | $60.05_{\pm3.32}$ | $10.78_{\pm7.79}$ | $0.10_{\pm1.00}$ | 0.01 | $40.00_{\pm24.49}$ | $0.13_{\pm0.08}$ |
| LEARNED2D-FLOW | $0.868_{\pm0.007}$ | $71.93_{\pm0.94}$ | $53.29_{\pm1.34}$ | $16.58_{\pm1.72}$ | $0.28_{\pm0.23}$ | 0.07 | $57.33_{\pm20.50}$ | $0.65_{\pm0.34}$ |
| I2B-FLOW | $0.910_{\pm0.005}$ | $97.61_{\pm0.11}$ | $97.20_{\pm0.11}$ | $97.24_{\pm0.66}$ | $54.88_{\pm0.26}$ | 91.83 | $51.56_{\pm1.01}$ | $45.11_{\pm1.20}$ |
| DIC-FLOW | $0.903_{\pm0.006}$ | $96.89_{\pm0.14}$ | $95.78_{\pm0.24}$ | $95.84_{\pm0.38}$ | $52.39_{\pm0.26}$ | 88.74 | $50.86_{\pm0.71}$ | $52.53_{\pm0.44}$ |
| TABFLOW | $0.914_{\pm0.002}$ | $95.03_{\pm0.04}$ | $92.87_{\pm0.04}$ | $77.55_{\pm0.19}$ | $\mathbf{61.94_{\pm0.53}}$ | 73.74 | $51.62_{\pm0.82}$ | $42.59_{\pm0.74}$ |
| TABREP-DDPM | $\mathbf{0.926_{\pm0.005}}$ | $\mathbf{98.97_{\pm0.10}}$ | $97.62_{\pm0.02}$ | $96.14_{\pm0.19}$ | $53.68_{\pm0.73}$ | $\mathbf{96.37}$ | $49.86_{\pm0.98}$ | $48.16_{\pm0.90}$ |
| TABREP-FLOW | $0.919_{\pm0.005}$ | $97.74_{\pm0.03}$ | $97.08_{\pm0.07}$ | $95.85_{\pm0.46}$ | $55.92_{\pm0.37}$ | 94.20 | $51.38_{\pm1.66}$ | $\mathbf{49.19_{\pm0.86}}$ |

**Stroke**

| Methods | AUC ↑ | CDE ↑ | PCC ↑ | $\alpha$ ↑ | $\beta$ ↑ | C2ST ↑ | MIA P. ↕ | MIA R.↕ |
|---|---|---|---|---|---|---|---|---|
| OneHot-DDPM | $0.797_{\pm0.033}$ | $98.74_{\pm0.11}$ | $97.65_{\pm0.10}$ | $96.78_{\pm1.08}$ | $56.27_{\pm0.22}$ | $98.62$ | $53.08_{\pm2.50}$ | $\mathbf{49.76_{\pm1.24}}$ |
| Learned1D-DDPM | $0.743_{\pm0.032}$ | $60.20_{\pm11.13}$ | $35.86_{\pm15.52}$ | $6.97_{\pm42.82}$ | $0.26_{\pm3.51}$ | $0.01$ | $15.00_{\pm0.20}$ | $0.33_{\pm10.00}$ |
| Learned2D-DDPM | $0.850_{\pm0.035}$ | $59.11_{\pm14.59}$ | $33.77_{\pm19.97}$ | $6.67_{\pm36.39}$ | $0.87_{\pm2.42}$ | $0.02$ | $18.33_{\pm0.33}$ | $0.49_{\pm13.02}$ |
| i2b-DDPM | $0.852_{\pm0.029}$ | $99.02_{\pm0.18}$ | $95.12_{\pm1.47}$ | $98.14_{\pm0.16}$ | $64.11_{\pm0.75}$ | $99.77$ | $50.72_{\pm2.05}$ | $49.11_{\pm1.59}$ |
| dic-DDPM | $0.824_{\pm0.021}$ | $98.98_{\pm0.09}$ | $\mathbf{98.00_{\pm2.13}}$ | $98.25_{\pm0.62}$ | $65.00_{\pm0.88}$ | $99.39$ | $52.97_{\pm1.19}$ | $52.03_{\pm1.76}$ |
| OneHot-Flow | $0.812_{\pm0.029}$ | $94.17_{\pm0.08}$ | $90.45_{\pm0.08}$ | $81.87_{\pm0.55}$ | $49.23_{\pm0.56}$ | $64.82$ | $49.32_{\pm1.47}$ | $39.51_{\pm1.25}$ |
| Learned1D-Flow | $0.142_{\pm0.033}$ | $75.48_{\pm6.55}$ | $56.60_{\pm9.05}$ | $71.63_{\pm35.72}$ | $0.45_{\pm0.13}$ | $0.16$ | $47.00_{\pm0.55}$ | $1.14_{\pm20.22}$ |
| Learned2D-Flow | $0.180_{\pm0.030}$ | $54.02_{\pm13.09}$ | $28.95_{\pm17.18}$ | $5.67_{\pm24.97}$ | $0.21_{\pm0.67}$ | $0.00$ | $20.00_{\pm0.16}$ | $0.16_{\pm20.00}$ |
| i2b-Flow | $0.797_{\pm0.027}$ | $98.72_{\pm0.11}$ | $94.65_{\pm1.33}$ | $98.26_{\pm0.27}$ | $65.23_{\pm0.26}$ | $97.19$ | $52.72_{\pm2.26}$ | $52.03_{\pm1.18}$ |
| dic-Flow | $0.807_{\pm0.019}$ | $98.50_{\pm0.23}$ | $92.71_{\pm2.35}$ | $97.76_{\pm0.64}$ | $65.77_{\pm1.95}$ | $97.33$ | $51.45_{\pm1.29}$ | $48.78_{\pm1.65}$ |
| TabFlow | $0.868_{\pm0.035}$ | $97.72_{\pm0.01}$ | $96.00_{\pm0.03}$ | $89.21_{\pm0.82}$ | $\mathbf{68.30_{\pm0.60}}$ | $89.19$ | $51.80_{\pm1.67}$ | $47.32_{\pm1.95}$ |
| TabRep-DDPM | $\mathbf{0.869_{\pm0.027}}$ | $\mathbf{99.14_{\pm0.20}}$ | $97.11_{\pm0.60}$ | $\mathbf{98.32_{\pm0.82}}$ | $57.17_{\pm0.77}$ | $\mathbf{100.00}$ | $51.74_{\pm1.85}$ | $50.89_{\pm0.93}$ |
| TabRep-Flow | $0.854_{\pm0.028}$ | $98.42_{\pm0.31}$ | $97.37_{\pm2.12}$ | $96.40_{\pm0.71}$ | $63.91_{\pm0.87}$ | $95.96$ | $\mathbf{50.66_{\pm1.77}}$ | $49.43_{\pm1.16}$ |

**Diabetes**

| Methods | F1 ↑ | CDE ↑ | PCC ↑ | $\alpha$ ↑ | $\beta$ ↑ | C2ST ↑ | MIA P. ↕ | MIA R.↕ |
|---|---|---|---|---|---|---|---|---|
| OneHot-DDPM | $0.363_{\pm0.008}$ | $69.22_{\pm0.04}$ | $50.14_{\pm0.06}$ | $9.93_{\pm0.16}$ | $2.34_{\pm0.11}$ | $1.74$ | $45.56_{\pm0.03}$ | $0.36_{\pm2.52}$ |
| Learned1D-DDPM | $0.179_{\pm0.010}$ | $63.40_{\pm4.59}$ | $39.94_{\pm5.34}$ | $0.00_{\pm0.00}$ | $0.00_{\pm0.00}$ | $0.00$ | $0.00_{\pm0.00}$ | $0.00_{\pm0.00}$ |
| Learned2D-DDPM | $0.205_{\pm0.007}$ | $64.30_{\pm1.31}$ | $41.49_{\pm1.74}$ | $0.01_{\pm0.01}$ | $0.00_{\pm0.00}$ | $0.00$ | $8.00_{\pm0.00}$ | $0.00_{\pm8.00}$ |
| i2b-DDPM | $0.370_{\pm0.008}$ | $99.38_{\pm0.01}$ | $98.84_{\pm0.03}$ | $97.70_{\pm0.13}$ | $46.83_{\pm0.39}$ | $93.98$ | $50.08_{\pm0.39}$ | $51.63_{\pm0.12}$ |
| dic-DDPM | $0.375_{\pm0.006}$ | $\mathbf{99.50_{\pm0.02}}$ | $\mathbf{99.12_{\pm0.01}}$ | $98.92_{\pm0.13}$ | $48.34_{\pm0.18}$ | $\mathbf{95.03}$ | $50.37_{\pm0.25}$ | $54.48_{\pm0.74}$ |
| OneHot-Flow | $0.372_{\pm0.005}$ | $96.65_{\pm0.03}$ | $94.61_{\pm1.99}$ | $97.43_{\pm0.06}$ | $41.64_{\pm0.16}$ | $55.44$ | $49.49_{\pm0.21}$ | $22.78_{\pm0.26}$ |
| Learned1D-Flow | $0.184_{\pm0.007}$ | $53.27_{\pm7.48}$ | $28.31_{\pm8.51}$ | $0.00_{\pm0.00}$ | $0.00_{\pm0.00}$ | $0.00$ | $0.00_{\pm0.00}$ | $0.00_{\pm0.00}$ |
| Learned2D-Flow | $0.177_{\pm0.008}$ | $68.39_{\pm9.23}$ | $46.71_{\pm10.95}$ | $0.00_{\pm0.00}$ | $0.00_{\pm0.00}$ | $0.00$ | $0.00_{\pm0.00}$ | $0.00_{\pm0.00}$ |
| i2b-Flow | $0.372_{\pm0.003}$ | $98.92_{\pm0.03}$ | $98.34_{\pm0.05}$ | $98.45_{\pm0.14}$ | $48.97_{\pm0.29}$ | $89.28$ | $49.86_{\pm0.26}$ | $48.25_{\pm0.13}$ |
| dic-Flow | $0.376_{\pm0.005}$ | $99.01_{\pm0.03}$ | $98.54_{\pm0.10}$ | $99.65_{\pm0.10}$ | $48.98_{\pm0.10}$ | $89.06$ | $50.10_{\pm0.12}$ | $48.01_{\pm0.18}$ |
| TabFlow | $0.376_{\pm0.006}$ | $98.04_{\pm0.02}$ | $96.82_{\pm0.01}$ | $79.60_{\pm0.09}$ | $\mathbf{51.58_{\pm0.04}}$ | $73.35$ | $50.19_{\pm0.39}$ | $44.14_{\pm0.15}$ |
| TabRep-DDPM | $0.373_{\pm0.003}$ | $99.36_{\pm0.02}$ | $98.75_{\pm0.03}$ | $97.19_{\pm0.22}$ | $46.19_{\pm0.49}$ | $92.65$ | $\mathbf{50.07_{\pm0.37}}$ | $51.43_{\pm0.13}$ |
| TabRep-Flow | $\mathbf{0.377_{\pm0.002}}$ | $99.00_{\pm0.02}$ | $98.46_{\pm0.05}$ | $\mathbf{99.08_{\pm0.13}}$ | $48.58_{\pm0.17}$ | $90.41$ | $50.24_{\pm0.37}$ | $\mathbf{50.33_{\pm0.13}}$ |

**Beijing**

| Methods | RMSE ↓ | CDE ↑ | PCC ↑ | $\alpha$ ↑ | $\beta$ ↑ | C2ST ↑ | MIA P. ↕ | MIA R.↕ |
|---|---|---|---|---|---|---|---|---|
| OneHot-DDPM | $2.143_{\pm0.339}$ | $74.21_{\pm0.07}$ | $63.68_{\pm0.05}$ | $48.88_{\pm0.03}$ | $19.07_{\pm0.13}$ | $19.68$ | $50.28_{\pm2.95}$ | $5.44_{\pm0.50}$ |
| Learned1D-DDPM | $0.921_{\pm0.006}$ | $79.49_{\pm3.50}$ | $62.23_{\pm5.01}$ | $26.94_{\pm27.91}$ | $8.49_{\pm5.95}$ | $13.51$ | $45.45_{\pm2.82}$ | $1.97_{\pm0.13}$ |
| Learned2D-DDPM | $0.969_{\pm0.005}$ | $81.89_{\pm1.24}$ | $66.02_{\pm2.37}$ | $79.49_{\pm16.01}$ | $7.73_{\pm1.53}$ | $55.83$ | $49.20_{\pm1.89}$ | $2.05_{\pm0.10}$ |
| i2b-DDPM | $0.542_{\pm0.008}$ | $98.66_{\pm0.04}$ | $96.95_{\pm0.21}$ | $97.92_{\pm0.15}$ | $59.27_{\pm0.14}$ | $94.28$ | $50.64_{\pm0.47}$ | $48.43_{\pm0.89}$ |
| dic-DDPM | $0.547_{\pm0.007}$ | $98.83_{\pm0.04}$ | $97.21_{\pm0.16}$ | $98.97_{\pm0.30}$ | $61.90_{\pm0.12}$ | $\mathbf{98.83}$ | $51.32_{\pm0.35}$ | $50.61_{\pm0.66}$ |
| OneHot-Flow | $0.765_{\pm0.016}$ | $84.61_{\pm0.02}$ | $67.28_{\pm4.64}$ | $84.38_{\pm0.61}$ | $20.32_{\pm0.19}$ | $35.76$ | $51.44_{\pm1.49}$ | $8.12_{\pm0.43}$ |
| Learned1D-Flow | $0.806_{\pm0.009}$ | $80.14_{\pm1.60}$ | $64.19_{\pm2.33}$ | $81.15_{\pm3.68}$ | $13.11_{\pm0.38}$ | $20.40$ | $44.02_{\pm3.58}$ | $1.13_{\pm0.09}$ |
| Learned2D-Flow | $0.787_{\pm0.007}$ | $79.50_{\pm0.85}$ | $63.04_{\pm1.01}$ | $43.00_{\pm4.90}$ | $7.55_{\pm4.05}$ | $14.58$ | $54.23_{\pm1.72}$ | $2.01_{\pm0.09}$ |
| i2b-Flow | $0.543_{\pm0.012}$ | $98.08_{\pm0.04}$ | $96.87_{\pm0.43}$ | $96.83_{\pm0.12}$ | $60.58_{\pm0.19}$ | $91.66$ | $49.64_{\pm0.47}$ | $45.77_{\pm1.07}$ |
| dic-Flow | $0.561_{\pm0.013}$ | $98.09_{\pm0.03}$ | $96.37_{\pm0.08}$ | $97.04_{\pm0.22}$ | $60.78_{\pm0.10}$ | $93.96$ | $50.86_{\pm0.71}$ | $52.53_{\pm0.44}$ |
| TabFlow | $0.574_{\pm0.01}$ | $96.44_{\pm0.06}$ | $93.71_{\pm0.07}$ | $94.81_{\pm0.42}$ | $59.47_{\pm0.28}$ | $87.23$ | $50.54_{\pm0.35}$ | $42.20_{\pm0.61}$ |
| TabRep-DDPM | $\mathbf{0.508_{\pm0.006}}$ | $\mathbf{99.11_{\pm0.03}}$ | $96.97_{\pm0.20}$ | $\mathbf{98.98_{\pm0.16}}$ | $\mathbf{64.08_{\pm0.18}}$ | $98.16$ | $51.02_{\pm0.42}$ | $\mathbf{49.50_{\pm0.52}}$ |
| TabRep-Flow | $0.536_{\pm0.006}$ | $98.28_{\pm0.07}$ | $96.92_{\pm0.21}$ | $98.16_{\pm0.13}$ | $62.65_{\pm0.12}$ | $92.26$ | $\mathbf{50.35_{\pm0.60}}$ | $49.14_{\pm0.81}$ |

**News**

| Methods | RMSE ↓ | CDE ↑ | PCC ↑ | $\alpha$ ↑ | $\beta$ ↑ | C2ST ↑ | MIA P. ↕ | MIA R.↕ |
|---|---|---|---|---|---|---|---|---|
| OneHot-DDPM | $0.840_{\pm0.02}$ | $98.11_{\pm0.06}$ | $92.78_{\pm0.08}$ | $96.33_{\pm0.24}$ | $46.87_{\pm0.28}$ | $95.80$ | $50.25_{\pm0.46}$ | $32.42_{\pm0.80}$ |
| Learned1D-DDPM | $0.858_{\pm0.01}$ | $96.45_{\pm0.14}$ | $95.00_{\pm0.32}$ | $90.48_{\pm9.11}$ | $19.54_{\pm4.84}$ | $21.58$ | $51.33_{\pm0.96}$ | $17.60_{\pm1.02}$ |
| Learned2D-DDPM | $0.857_{\pm0.023}$ | $96.06_{\pm0.49}$ | $94.49_{\pm0.84}$ | $88.83_{\pm4.52}$ | $5.11_{\pm5.24}$ | $20.41$ | $50.85_{\pm2.97}$ | $3.83_{\pm0.39}$ |
| i2b-DDPM | $0.844_{\pm0.013}$ | $98.41_{\pm0.02}$ | $98.57_{\pm0.18}$ | $95.02_{\pm0.10}$ | $48.22_{\pm0.18}$ | $96.07$ | $50.88_{\pm0.57}$ | $39.82_{\pm0.93}$ |
| dic-DDPM | $0.866_{\pm0.019}$ | $98.22_{\pm0.06}$ | $98.50_{\pm0.36}$ | $97.08_{\pm0.25}$ | $47.88_{\pm0.10}$ | $95.75$ | $49.54_{\pm0.57}$ | $39.80_{\pm0.35}$ |
| OneHot-Flow | $0.850_{\pm0.017}$ | $96.27_{\pm0.05}$ | $98.11_{\pm0.02}$ | $\mathbf{97.78_{\pm0.13}}$ | $43.06_{\pm0.62}$ | $84.56$ | $50.45_{\pm2.02}$ | $14.84_{\pm0.65}$ |
| Learned1D-Flow | $0.873_{\pm0.007}$ | $95.07_{\pm0.14}$ | $95.07_{\pm0.12}$ | $89.16_{\pm5.89}$ | $18.17_{\pm6.18}$ | $79.17$ | $49.85_{\pm1.69}$ | $10.28_{\pm0.47}$ |
| Learned2D-Flow | $0.866_{\pm0.005}$ | $94.85_{\pm0.38}$ | $94.99_{\pm0.89}$ | $80.66_{\pm15.23}$ | $14.63_{\pm4.14}$ | $78.96$ | $49.59_{\pm1.54}$ | $14.19_{\pm0.69}$ |
| i2b-Flow | $0.847_{\pm0.014}$ | $96.64_{\pm0.05}$ | $98.38_{\pm0.34}$ | $88.39_{\pm0.11}$ | $\mathbf{51.85_{\pm0.24}}$ | $89.47$ | $51.12_{\pm0.71}$ | $36.35_{\pm0.53}$ |
| dic-Flow | $0.853_{\pm0.014}$ | $96.58_{\pm0.04}$ | $97.49_{\pm0.34}$ | $92.28_{\pm0.25}$ | $50.79_{\pm0.29}$ | $88.30$ | $50.43_{\pm0.63}$ | $37.56_{\pm0.77}$ |
| TabFlow | $0.850_{\pm0.017}$ | $96.51_{\pm0.08}$ | $97.93_{\pm0.02}$ | $92.68_{\pm0.31}$ | $50.08_{\pm0.26}$ | $87.33$ | $51.01_{\pm0.24}$ | $28.00_{\pm0.89}$ |
| TabRep-DDPM | $0.836_{\pm0.001}$ | $\mathbf{98.46_{\pm0.01}}$ | $99.09_{\pm0.05}$ | $95.35_{\pm0.11}$ | $48.49_{\pm0.12}$ | $\mathbf{96.70}$ | $\mathbf{49.87_{\pm0.99}}$ | $40.10_{\pm0.55}$ |
| TabRep-Flow | $\mathbf{0.814_{\pm0.002}}$ | $96.89_{\pm0.03}$ | $98.34_{\pm0.29}$ | $90.91_{\pm0.25}$ | $51.75_{\pm0.16}$ | $88.13$ | $50.90_{\pm0.93}$ | $35.48_{\pm0.78}$ |

### D.3 Additional Training and Sampling Duration Results

We conducted an additional Training and Sampling Duration experiment on the largest dataset among our dataset suite (Diabetes dataset) with $99,473$ samples, 8 numerical features, and 21 categorical features. As observed in Table 13, we save around 1700 seconds compared to TabDDPM and around 4900 seconds compared to TabSYN during training and sampling.

Table 13: Training and Sampling Duration in Seconds.

| METHODS | TRAINING | SAMPLING | TOTAL |
|---|---|---|---|
| TABDDPM | 3455 | 268 | 3723 |
| TABSYN | 6003 + 882 | 18 | 6903 |
| TABREP-DDPM | 1980 | 114 | 2094 |
| TABREP-FLOW | **2002** | **7** | **2009** |

### D.4 Additional Results on High Cardinality and Imbalanced Toy Datasets

**High Cardinality**. We curate synthetic toy datasets of high cardinality categorical variables and imbalanced datasets to reinforce our generalizability claims. Our high cardinality toy dataset is a regression task with two categorical features. The first categorical feature is of high cardinality, with 1000 unique categories where each category is assigned a base effect drawn from a normal distribution. The other categorical feature may take 3 values, each having fixed effects of $1.0$, $-1.0$, and $0.5$ respectively. The target label is computed by summing the base effect from the high-cardinality category, the fixed effect from the other categorical feature, and an independent numerical feature drawn from a standard normal distribution. Additional Gaussian noise is added to perturb the data.

Table 14: Performance on High Cardinality Setting.

| | RMSE ↓ | CDE ↑ | PWC ↑ | C2ST ↑ | $\alpha$-PRECISION ↑ | $\beta$-RECALL ↑ |
|---|---|---|---|---|---|---|
| TABDDPM | $0.8253_{\pm 0.1419}$ | $42.93_{\pm 0.17}$ | $11.20_{\pm 0.09}$ | $0.41_{\pm 0.02}$ | $0.46_{\pm 0.01}$ | $0.02_{\pm 0.01}$ |
| TABSYN | $0.4775_{\pm 0.0129}$ | $94.23_{\pm 0.56}$ | $\mathbf{78.37_{\pm 2.00}}$ | $\mathbf{100.00_{\pm 0.00}}$ | $99.13_{\pm 0.31}$ | $35.35_{\pm 0.47}$ |
| TABREP-DDPM | $\mathbf{0.4662_{\pm 0.0071}}$ | $80.88_{\pm 0.15}$ | $59.96_{\pm 1.08}$ | $25.45_{\pm 0.10}$ | $99.31_{\pm 0.21}$ | $16.47_{\pm 0.32}$ |
| TABREP-FLOW | $0.4812_{\pm 0.0104}$ | $\mathbf{94.38_{\pm 0.28}}$ | $76.84_{\pm 1.08}$ | $98.59_{\pm 0.94}$ | $\mathbf{99.32_{\pm 0.22}}$ | $\mathbf{36.00_{\pm 0.24}}$ |

As shown in Table 14, it is worth noting that DDPM models, including TabRep-DDPM and TabDDPM, perform poorly in CDE, PWC, and C2ST tasks, yet are able to model RMSE and $\alpha$-precision well. This indicates that with high cardinality, TabRep-DDPM is less capable of learning conditional distributions across features. In contrast, our proposed TabRep-Flow performs on par with TabSYN, as flow-matching models' smooth differentiable transformation allows them to capture subtle conditional variations, and TabSYN's latent space allows for the learning of a simpler latent distribution.

**Dataset Imbalance**. The imbalanced toy dataset is a regression task with one binary categorical feature (distributed 95% class A, 5% class B). Each row also contains a numeric feature drawn from a standard normal distribution. The target is constructed by applying a category-specific linear function before addition of some Gaussian noise to generate variations in the data.

Table 15: Performance on Imbalanced Setting.

| | RMSE ↓ | CDE ↑ | PWC ↑ | C2ST ↑ | $\alpha$-PRECISION ↑ | $\beta$-RECALL ↑ |
|---|---|---|---|---|---|---|
| TABDDPM | $0.1694_{\pm 0.0016}$ | $98.93_{\pm 0.54}$ | $95.87_{\pm 5.43}$ | $98.98_{\pm 1.57}$ | $98.96_{\pm 0.68}$ | $50.29_{\pm 0.49}$ |
| TABSYN | $0.1708_{\pm 0.0018}$ | $94.98_{\pm 0.04}$ | $96.36_{\pm 0.18}$ | $90.27_{\pm 0.65}$ | $95.84_{\pm 1.13}$ | $48.89_{\pm 0.23}$ |
| TABREP-DDPM | $\mathbf{0.1688_{\pm 0.0010}}$ | $\mathbf{99.44_{\pm 0.13}}$ | $\mathbf{96.82_{\pm 3.97}}$ | $\mathbf{99.42_{\pm 0.54}}$ | $\mathbf{99.46_{\pm 0.08}}$ | $50.96_{\pm 0.77}$ |
| TABREP-FLOW | $0.1689_{\pm 0.0025}$ | $98.52_{\pm 0.12}$ | $90.49_{\pm 5.36}$ | $99.06_{\pm 0.47}$ | $96.69_{\pm 0.28}$ | $\mathbf{51.30_{\pm 0.31}}$ |

In Table 15, we see that for imbalanced data, our proposed methods achieve results that are better compared to existing models like TabDDPM and TabSYN, showing that our methods are generalizable to cases where training data may be highly imbalanced.

### D.5 TabSYN's Latent Representation Dimension

By default, TabSYN has a latent dimensionality of 4. To address the concern regarding TabSYN's dimensionality, we run TabSYN using the same dimensions (2D latent space) as our TabRep representation. As observed in Table 16, TabSYN with a 2D latent dimension performs much worse than TabSYN with a 4D latent dimension.

Table 16: AUC (classification) and RMSE (regression) scores of Machine Learning Efficiency. Higher scores indicate better performance.

| | AUC ↑ | | | | F1 ↑ | RMSE ↓ | |
|---|---|---|---|---|---|---|---|
| METHODS | ADULT | DEFAULT | SHOPPERS | STROKE | DIABETES | BEIJING | NEWS |
| TabSYN | $0.906_{\pm.001}$ | $0.755_{\pm.004}$ | $0.918_{\pm.004}$ | $0.845_{\pm.035}$ | $0.361_{\pm.001}$ | $0.586_{\pm.013}$ | $0.862_{\pm.021}$ |
| TabSYN (2D Latent Space) | $0.892_{\pm.002}$ | $0.752_{\pm.005}$ | $0.916_{\pm.002}$ | $0.811_{\pm.032}$ | $0.368_{\pm.002}$ | $0.720_{\pm.015}$ | $0.868_{\pm.003}$ |
| TabRep-DDPM | $\mathbf{0.913}_{\pm.002}$ | $0.764_{\pm.005}$ | $\mathbf{0.926}_{\pm.005}$ | $\mathbf{0.869}_{\pm.027}$ | $0.373_{\pm.003}$ | $\mathbf{0.508}_{\pm.006}$ | $0.836_{\pm.001}$ |
| TabRep-Flow | $0.912_{\pm.002}$ | $\mathbf{0.782}_{\pm.005}$ | $0.919_{\pm.005}$ | $0.830_{\pm.028}$ | $\mathbf{0.377}_{\pm.002}$ | $0.536_{\pm.006}$ | $\mathbf{0.814}_{\pm.002}$ |

In terms of computational costs on the Adult dataset, Table 17 highlights that TabSYN in a 2D Latent Space saves close to 500 seconds while compromising on accuracy when compared to vanilla TabSYN (4D Latent Space). However, it still consumes around 1000 seconds extra when compared to TabRep methods.

Table 17: Training and Sampling Duration in Seconds.

| METHODS | TRAINING | SAMPLING | TOTAL |
|---|---|---|---|
| TabSYN | 2374 + 1085 | 11 | 3470 |
| TabSYN (2D Latent Space) | 2333 + 670 | 6 | 3009 |
| TabRep-DDPM | 2071 | 59 | 2130 |
| TabRep-Flow | **2028** | **3** | **2031** |

