# OpenReview forum: "TabRep: Training Tabular Diffusion Models with a Simple and Effective Continuous Representation"
_TMLR — Accepted by TMLR_

### Review · Reviewer_EpAf · 2025-10-06

**Summary Of Contributions:**

### Summary
The paper introduces **TabRep**, a simple yet effective framework for training tabular diffusion models using a unified continuous data representation rather than separate modeling of heterogeneous data. The core idea is a new encoding scheme called **CatConverter**, which maps categorical variables to a dense 2-D representation on the unit circle inspired by the discrete Fourier transform. The authors provide insights into why conventional sparse encodings (e.g., one-hot) create singular regions that increase the variance of score estimates and harm learning. By contrast, the CatConverter representation maintains both density and separability, avoiding such issues while being computationally light. The paper supports these ideas through theoretical analysis and extensive experiments on seven benchmark tabular datasets, showing consistent improvements in generation quality, privacy preservation, and computational efficiency over prior diffusion-based tabular generators such as STaSy, CoDi, TabDDPM, TabSYN, and TabDiff.

### Strengths
TabRep offers a clear and well-motivated insight into the geometric limitations of one-hot encodings in diffusion models and introduces an elegant, mathematically grounded continuous representation that is easily implemented and model-agnostic. The empirical results are comprehensive, including ablations, privacy analysis, and comparisons across both DDPM and flow-matching variants.

### Weaknesses
The theoretical component, while interesting, remains largely heuristic, i.e., there is no formal empirical validation linking the proposed variance analysis directly to training dynamics. Moreover, the improvements, though consistent, are sometimes modest, and the approach may be limited to datasets with relatively small categorical cardinalities (the paper only validates up to 128 categories). Finally, the broader generalization beyond diffusion-based methods or more complex relational/tabular settings (e.g., multi-table or time-series) is unexplored.

**Audience:**

Yes

**Audience Explanation:**

Since the paper addresses a practical and long-standing problem in tabular generative modeling, an area relevant to both applied machine learning and generative model researchers. The proposed continuous representation is conceptually simple and may influence how future diffusion-based models handle mixed-type data.

**Broader Impact Concerns:**

The work aims to generate synthetic tabular data that preserve statistical fidelity while maintaining privacy, which aligns with positive social and ethical objectives. Nevertheless, synthetic data can still leak sensitive information if membership inference defenses are insufficient or if generated data are misused for re-identification. While the authors include MIA results showing near-random recall, a more comprehensive privacy analysis, e.g., evaluating memorization or semantic leakage, would strengthen the ethical assessment. Adding a clear statement cautioning against deploying TabRep-generated data without domain-specific privacy auditing would appropriately address these broader impact concerns.

**Claims And Evidence:**

Yes

**Claims Explanation:**

I think the claims in the submission are mostly accurate and clear.

The major claims are 1) the proposed representation improves training stability and generative quality, 2) it preserves privacy comparably to baselines, and 3) it reduces computational cost which are supported by extensive experiments and clearly reported metrics. The ablations on encoding schemes (Table 3) convincingly show that CatConverter is a superior representation for both DDPM and flow-matching models. However, the theoretical claims regarding singular hyperplanes and score-variance scaling are only partially validated; while mathematically correct, they are not empirically connected to observed learning behavior. Overall, the empirical evidence is sound and reproducible, though the theoretical grounding could be better substantiated.

**Requested Changes:**

The following changes would strengthen the paper:

1. Provide clearer empirical evidence linking the proposed geometric analysis to actual training stability or loss-variance behavior, e.g., by visualizing or quantifying how CatConverter affects gradient variance or score consistency.
2. Discuss the limitations of CatConverter for very high-cardinality categorical features or mixed hierarchical schemas; current results (up to K = 128) do not demonstrate scalability.
3. Expand on potential extensions to multi-table or relational data settings.
4. Improve clarity of notation and equation numbering (especially in Section 4.1).
5. Include runtime comparisons normalized by hardware specifications and add a brief discussion on hyperparameter sensitivity.

---

> ### Author Response · Authors · 2025-10-26
> **Rebuttal**
>
> We sincerely thank the reviewer for all the detailed feedback. We have incorporated the feedback into our newly revised manuscript. Please find all changes that are highlighted in **blue**.
>
> > Provide clearer empirical evidence linking the proposed geometric analysis to actual training stability or loss-variance behavior, e.g., by visualizing or quantifying how CatConverter affects gradient variance or score consistency.
> >
>
> We measure training loss by incrementally training TabRep’s diffusion models across all seven datasets, comparing the performance of CatConverter and one-hot encoding. Across the majority of datasets, CatConverter demonstrates both faster convergence, lower loss values, and a reduction in the variability of the loss.
>
> - Page 30, Appendix D.5, Loss Stability of CatConverter vs. One-Hot Encoding, Figure 7
>
> > Discuss the limitations of CatConverter for very high-cardinality categorical features or mixed hierarchical schemas; current results (up to K = 128) do not demonstrate scalability.
> >
>
> In Appendix D.4 of our paper, we curate synthetic toy datasets of high cardinality categorical variables (1000 unique categories) and conducted experiments. “As shown in Table 14, it is worth noting that DDPM models including TabRep-DDPM and TabDDPM perform poorly in CDE, PWC, and C2ST tasks, yet are able to model RMSE and $\alpha$-precision well. This indicates that with high cardinality, TabRep-DDPM is less capable of learning conditional distributions across features. In contrast, our proposed TabRep-Flow performs on par with TabSYN, as flow-matching models’ smooth differentiable transformation allows them to capture subtle conditional variations, and TabSYN’s latent space allows for the learning of a simpler latent distribution. “
>
> - Page 29, Appendix D.4
>
> > Expand on potential extensions to multi-table or relational data settings.
> >
>
> We understand that there are works such as RelDiff, ClavaDDPM, TabARGN, and REaLTabFormer [1,2,3,4] that focus on multi-table and relational data. For instance, RelDiff employs graphical neural network techniques to capture the correlation between different tables/relational data. However, our work does not specifically address the main challenges of relational data, such as capturing their complex structural hierarchies and dependencies across their interconnected table. Therefore, we have left the exploration of relational data to future work.
>
> [1] Hudovernik, Valter, et al. "RelDiff: Relational Data Generative Modeling with Graph-Based Diffusion Models."
>
> [2] Pang, Wei, et al. "Clavaddpm: Multi-relational data synthesis with cluster-guided diffusion models."
>
> [3] Tiwald, Paul, et al. "Tabularargn: A flexible and efficient auto-regressive framework for generating high-fidelity synthetic data."
>
> [4] Solatorio, Aivin V., and Olivier Dupriez. "Realtabformer: Generating realistic relational and tabular data using transformers."
>
> > Improve clarity of notation and equation numbering (especially in Section 4.1).
> >
>
> We added to Section 4.1 in Pages 5,6, and 7 to help clarify notation, equation numbering, and our method.
>
> - Pages 5,6, and 7, Section 4.1
>
> > Include runtime comparisons normalized by hardware specifications and add a brief discussion on hyperparameter sensitivity.
> >
>
> We included runtime comparisons in Tables 7 and 13, Pages 11 and 29, which are normalized by hardware specifications in Appendix B Implementation, Page 19. We also added further discussion on hyperparameter sensitivity in Appendix B.3 Hyperparameters Page 21.
>
> - Runtime: Tables 7 and 13, Pages 11 and 29
> - Hardware: Appendix B Implementation, Page 19
> - Hyperparameters: Appendix B.3, Page 21
>
> > The work aims to generate synthetic tabular data that preserve statistical fidelity while maintaining privacy, which aligns with positive social and ethical objectives. Nevertheless, synthetic data can still leak sensitive information if membership inference defenses are insufficient or if generated data are misused for re-identification. While the authors include MIA results showing near-random recall, a more comprehensive privacy analysis, e.g., evaluating memorization or semantic leakage, would strengthen the ethical assessment. Adding a clear statement cautioning against deploying TabRep-generated data without domain-specific privacy auditing would appropriately address these broader impact concerns.
> >
>
> We added a “Declarations” section before our References to address this.
>
> - Page 12, Declarations, “Broader Impact”

---

### Review · Reviewer_XRab · 2025-10-10

**Summary Of Contributions:**

This paper proposes TabRep that applies diffusion models for tabular data generation. The main idea is to convert discrete attributes to continuous ones using CatConverter() and then train a diffusion model or flow matching for generating tabular data.

**Audience:**

Yes

**Audience Explanation:**

- Tabular data generation is an interesting and impactful problem.

- The proposed technique to covert discrete to continuous features is quite interesting.

**Broader Impact Concerns:**

There are no broader impact concerns.

**Claims And Evidence:**

Yes

**Claims Explanation:**

- In this paper, it requires to deal with both continuous and discrete attributes in TabRep. The authors proposes CatConverter() for converting discrete attributes to the continuous ones. They developed some theories to explain why it is not a good idea of using one-hot vectors to represent the discrete values.

- The CatConverter makes sense and supports TabRep to learn from discrete feature more efficiently.

**Requested Changes:**

-  Geometric Implications on the Data Representation is currently hard to understand completely regarding its theoretical meaning and impact to concrete attributes. For example, in Theorem 4.1, why we need to consider the forward process: $q_t(x \mid e_k)$. It is beneficial if the authors give more explanation over this and further articulate the problem setting here, i.e., how the one-hot vectors are relevant to the setting.

-  It is unclear how to convert back two continuous values for a discrete attribute to a discrete value in a generated row because these two continuous values might not exactly match the (sin, cos) of the CatConverter().

- The authors need to further articulate QuantileTransformer in Line 5 of Alg. 1

- Currently, there are many works using Large Language Models for tabular data generation. The authors need to compare to these works or at least discuss the pros and cons of two approaches (i.e., using diffusion models and LLMs for tabular data generation).

---

> ### Author Response · Authors · 2025-10-26
> **Rebuttal**
>
> We sincerely thank the reviewer for all the detailed feedback. We have incorporated the feedback into our newly revised manuscript. Please find all changes that are highlighted in **blue**.
>
> > Geometric Implications on the Data Representation is currently hard to understand completely regarding its theoretical meaning and impact to concrete attributes. For example, in Theorem 4.1, why we need to consider the forward process: . It is beneficial if the authors give more explanation over this and further articulate the problem setting here, i.e., how the one-hot vectors are relevant to the setting.
> >
>
> We elaborate further and hope to clarify the “Geometric Implications on the Data Representation”
>
> - Page 5
>
> > It is unclear how to convert back two continuous values for a discrete attribute to a discrete value in a generated row because these two continuous values might not exactly match the (sin, cos) of the CatConverter().
> >
>
> We added a section titled “InvCatConverter” to clarify the conversion.
>
> - Page 7, “InvCatConverter”
>
> > The authors need to further articulate QuantileTransformer in Line 5 of Alg. 1
> >
>
> We add further explanations regarding the QuantileTransformer in the Appendix.
>
> - Page 21, B.2 Architecture, QuantileTransformer
>
> > Currently, there are many works using Large Language Models for tabular data generation. The authors need to compare to these works or at least discuss the pros and cons of two approaches (i.e., using diffusion models and LLMs for tabular data generation).
> >
>
> We added to our related works regarding LLMs for tabular data generation as well as comparisons, including pros and cons.
>
> - Pages 2 and 3, Related Works.

---

### Review · Reviewer_7iCn · 2025-10-16

**Summary Of Contributions:**

This paper introduces TabRep, a method for tabular data generation featuring a unified, continuous representation for mixed-type data. The core contribution is CatConverter, a novel encoder that maps categorical features to a dense, two-dimensional circular space. This design is motivated by a geometric analysis showing that common sparse representations create "singular hyperplanes" that impede diffusion model training.

Strengths: The representation is novel and well-motivated, supported by strong empirical results across many benchmarks, and validated with thorough ablation studies. The paper is exceptionally clear.

Weaknesses: The method relies on a heuristic ordering for ordinal features and uses a simple casting method for out-of-index values which may introduce bias.

Overall, this is a high-quality paper making a significant contribution to generative modeling for tabular data.

**Audience:**

Yes

**Audience Explanation:**

The paper will interest the TMLR audience due to its focus on generative modeling for tabular data, a common and important data type. The proposed solution is novel yet simple, efficient, and pushes state-of-the-art performance, making it highly relevant to both researchers and practitioners.

**Broader Impact Concerns:**

I do not have any broader impact concerns with this work. The paper focuses on a foundational machine learning technique with positive societal implications for data privacy and augmentation. The authors do not need to add a Broader Impact Statement.

**Claims And Evidence:**

Yes

**Claims Explanation:**

The claims are well-supported. The paper provides a strong theoretical motivation for its continuous representation (Theorem 4.1), backs it up with comprehensive experiments on diverse datasets against relevant baselines, and uses strong ablation studies to validate its specific design choices. The results convincingly demonstrate the method's effectiveness.

**Requested Changes:**

The paper is in excellent shape. The following suggestions would further strengthen it:

1. Expand discussion on feature ordering: The performance on ordinal features depends on the input ordering (lexicographical vs. random, as shown in Appendix B.2). It would strengthen the paper to add a brief discussion in the main text about this sensitivity and mention the potential for future work to explore more principled or learnable ordering strategies.

2. Briefly analyze the impact of OOI casting: The paper transparently reports the out-of-index (OOI) casting rate in the appendix. A short discussion on the potential implications of this—for example, whether this disproportionately inflates the frequency of the 0-th category and how that might impact downstream tasks, especially in cases of class imbalance—would add more depth to the analysis.

---

> ### Author Response · Authors · 2025-10-26
> **Rebuttal**
>
> We sincerely thank the reviewer for all the detailed feedback. We have incorporated the feedback into our newly revised manuscript. Please find all changes that are highlighted in **blue**.
>
> > Expand discussion on feature ordering: The performance on ordinal features depends on the input ordering (lexicographical vs. random, as shown in Appendix B.2). It would strengthen the paper to add a brief discussion in the main text about this sensitivity and mention the potential for future work to explore more principled or learnable ordering strategies.
> >
>
> We have added a brief discussion in the main text about the feature ordering sensitivity as well as the potential for future work to explore more principled or learnable ordering strategies
>
> - Pages 9 and 10, “Ordering of Categorical Feature”.
>
> > Briefly analyze the impact of OOI casting: The paper transparently reports the out-of-index (OOI) casting rate in the appendix. A short discussion on the potential implications of this—for example, whether this disproportionately inflates the frequency of the 0-th category and how that might impact downstream tasks, especially in cases of class imbalance—would add more depth to the analysis.
> >
>
> We include more analysis on the impact of OOI casting
>
> - Pages 20 and 21, B.2 Architecture, “Out-of-index (OOI)”.

---

### Author Response · Authors · 2025-10-26
**Summary Response**

We sincerely thank the reviewers for their constructive feedback. We are encouraged that reviewers appreciated the ***novel and interesting*** (**7iCn, XRab**) *approach* as well as the **strong and comprehensive empirical results** to tackle tabular data generation.

Our work addresses the fundamental challenge of generating tabular data under a unified data representation. Specifically:

- We introduce geometric insights to understand the properties of a continuous data manifold for tabular diffusion models.
- We craft an abstract and useful representation, that enables diffusion models to capture the posterior distribution efficaciously.
- We conduct comprehensive experiments against tabular diffusion baselines across various datasets and benchmarks. The results showcase that TabRep consistently outperforms these baselines in quality, privacy preservation, and computation cost, indicating our superior capabilities to generate tabular data.

We provide detailed responses to individual reviewer comments and questions below. We have incorporated the feedback into our newly revised manuscript. Please find all changes that are highlighted in **blue**.

---

### Decision · Action_Editor_rvVV · 2025-12-04

**Recommendation:** Accept with minor revision

**Additional Comments:**

- **Notation Fix**: Remove redundant superscript (i) from $K_j^{(i)}$ (Equations 11-12, Section 4.2, possibly elsewhere). As the AE understands the text, cardinality is a column property and not row-varying.

- **Optional improvement**: Consider moving loss stability analysis (Appendix D.5, Figure 7) to main text to strengthen theory-practice connection.

- **Clarify ordering**: Explicitly state in Section "Ordering of Categorical Feature" (pp. 9-10) that lexicographic ordering is a heuristic baseline, not optimized for semantic structure.

**Audience:**

Yes

**Audience Explanation:**

Tabular data generation is practically important for data augmentation, privacy-preserving ML, and imbalanced datasets. The work advances tabular diffusion models with a simple, effective encoding that is "plug-and-play" for DDPM and Flow Matching. All three reviewers agreed on relevance. The experimental evaluation (7 datasets, 5 baselines, multiple metrics) and honest limitation reporting provide value for researchers and practitioners in generative modeling and tabular ML.

**Claims And Evidence:**

Yes

**Claims Explanation:**

The AE finds that the core claims are adequately supported:

- *Main claim (CatConverter improves tabular diffusion models)*: Ablation studies (Table 3) demonstrate consistent improvements over one-hot, learned embeddings, and Analog Bits across 7 datasets. The revised training loss analysis (Appendix D.5, Figure 7) shows CatConverter achieves faster convergence and lower loss than one-hot encoding.

- *Theoretical motivation (singular hyperplanes)*: Theorem 4.1 is mathematically correct and provides geometric intuition for why sparse representations are problematic. However, the causal link between theory and observed improvements remains somewhat heuristic, as Reviewer EpAf noted.

- *Performance claims*: Results consistently show TabRep matches or exceeds baselines on quality, privacy, and efficiency. The claim of "first to exceed real data quality" holds for some datasets (Default, Stroke) but not all (News), which authors acknowledge.

- *Limitations*: Authors honestly report constraints, including struggles with $K=1000$ categories (TabRep-DDPM) and feature ordering sensitivity.

- *Minor gaps*: Only lexicographic vs. random ordering tested (not semantic); performance variation across datasets unexplained; LLM discussion is a bit superficial.

Overall, claims are reasonably supported for solid empirical work.